# Entorhinal cortex directs learning-related changes in CA1 representations

Christine Grienberger[1,2] & Jeffrey C. Magee[1✉]

Learning-related changes in brain activity are thought to underlie adaptive behaviours[1,2]. For instance, the learning of a reward site by rodents requires the development of an over-representation of that location in the hippocampus[3–6]. How this learning-related change occurs remains unknown. Here we recorded hippocampal CA1 population activity as mice learned a reward location on a linear treadmill. Physiological and pharmacological evidence suggests that the adaptive over-representation required behavioural timescale synaptic plasticity (BTSP)[7]. BTSP is known to be driven by dendritic voltage signals that we proposed were initiated by input from entorhinal cortex layer 3 (EC3). Accordingly, the CA1 over-representation was largely removed by optogenetic inhibition of EC3 activity. Recordings from EC3 neurons revealed an activity pattern that could provide an instructive signal directing BTSP to generate the over-representation. Consistent with this function, our observations show that exposure to a second environment possessing a prominent reward-predictive cue resulted in both EC3 activity and CA1 place field density that were more elevated at the cue than at the reward. These data indicate that learning-related changes in the hippocampus are produced by synaptic plasticity directed by an instructive signal from the EC3 that seems to be specifically adapted to the behaviourally relevant features of the environment.

The behavioural experience of animals has been found to shape population activity in the hippocampus, and this experience-dependent neuronal activity is required to learn rewarded locations[3–6]. Such learning-related neuronal changes are commonly thought to be mediated by synaptic plasticity, generally of the Hebbian type[8–10]. To directly examine the physiological processes by which experience alters hippocampal population activity, we used two-photon Ca²⁺ imaging to record the activity of GCaMP6f-expressing dorsal CA1 pyramidal neurons[11] in head-fixed mice engaged in a spatial learning task (Fig. 1a). The task consisted of two phases. Mice were first habituated to the linear track treadmill using a blank belt of 180 cm length, with the reward (10% sucrose solution) location varying from lap to lap (Fig. 1b–k). On day 0 (the final day of this habituation phase), the animals' lick rates and running speeds were uniform throughout the environment (Fig. 1b–f), and CA1 place cells evenly tiled the space (Fig. 1g,h). In the second phase, the reward was delivered at a single fixed location, and the track contained several sensory cues uniformly distributed in space (day 1: first exposure to the fixed reward location; Fig. 1b–l). During this session, the animals gradually restricted their licking to the part of the environment around the reward (Fig. 1b–d) and concurrently slowed their running speed when approaching the reward delivery site (Fig. 1e,f). In parallel to these behavioural changes, we observed an increase in the total number of CA1 place cells, with the density of place cells near the reward site elevated over twofold[3,4,6] (Fig. 1g–i). Spatial information content (Fig. 1j) and lap-to-lap reliability (Fig. 1k) of individual place fields were also enhanced. Finally, the place cell

population vector correlation was significantly lower when compared between days versus within days (Fig. 1k). Together, these results indicate that the learning of the reward location on day 1 is associated with an alteration in the CA1 representation that includes a strongly elevated place cell density near the reward, the presence of which is significantly correlated with low running speeds measured around the rewarded location (Fig. 1l). This so-called reward over-representation is similar to the CA1 activity adaptations previously found to be required for the successful learning of the reward location[5].

## Role of BTSP

Next we questioned whether a recently discovered synaptic plasticity type, BTSP[12,13], could underlie the experience-dependent formation of the CA1 representation on day 1. BTSP is exclusively driven by long-duration dendritic voltage signals, Ca²⁺ plateau potentials ('plateaus'), that can induce plasticity in a single trial[14–18] (Fig. 2a, left and middle). Moreover, BTSP follows an asymmetric learning rule that operates on the behaviourally relevant timescale of seconds. Therefore, it produces predictive neuronal activity; that is, the firing field generated by BTSP precedes the location of the plateau by an amount that depends on the running speed (Fig. 2a, left and middle). Another effect of the BTSP timescale is that there is a direct relationship between the animal's running speed in the lap in which the place field first appeared ('induction lap') and the width of the resulting place field in space (Fig. 2a, right). Consistent with the involvement

[1]Howard Hughes Medical Institute, Baylor College of Medicine, Houston, TX, USA. [2]Present address: Brandeis University, Department of Biology and Volen National Center for Complex Systems, Waltham, MA, USA. ✉e-mail: jcmagee@bcm.edu

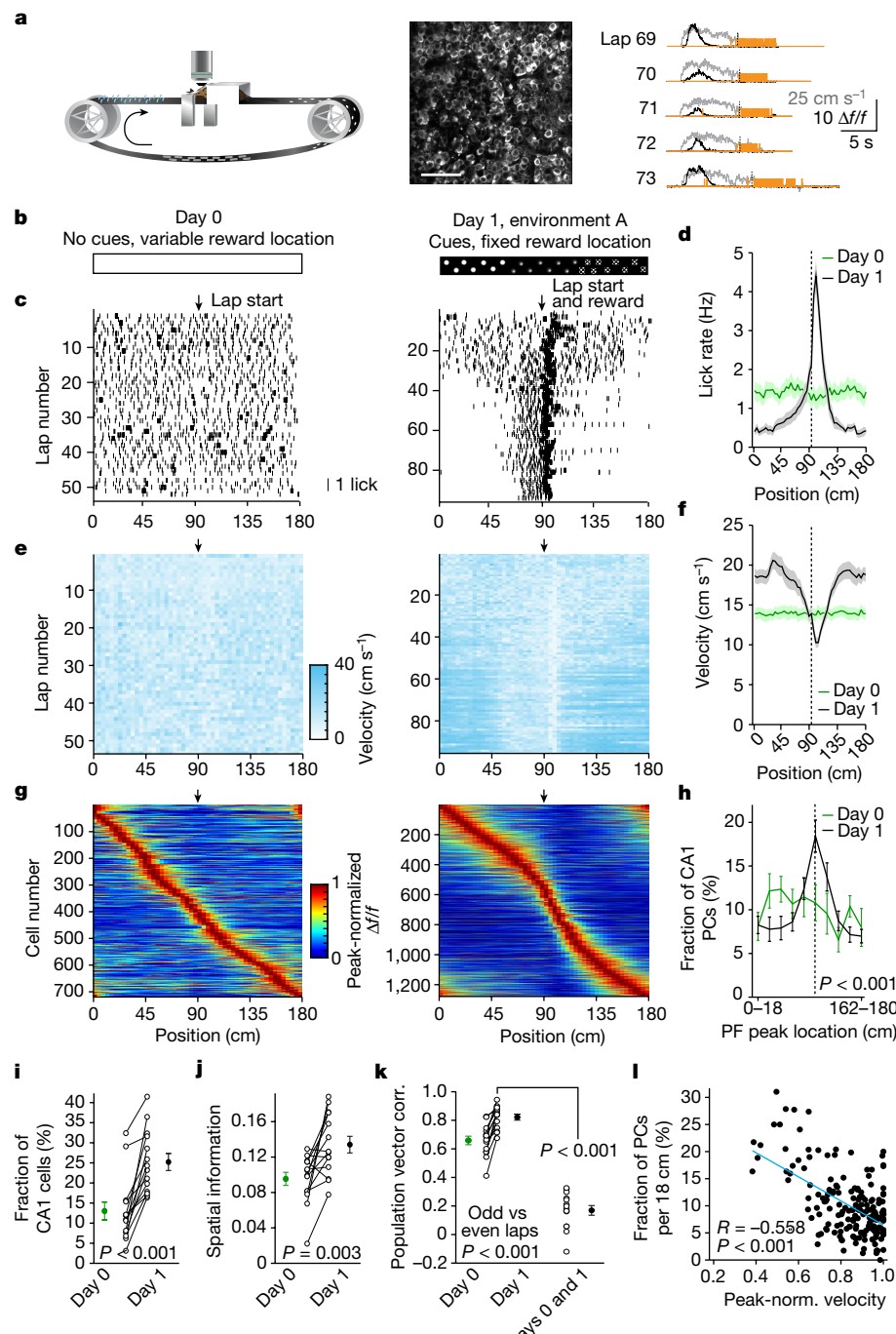

**Fig. 1 | Experience-dependent changes in CA1 representations. a,** Left: the experimental setup in which a mouse learns the location of a water reward. Middle: a representative time-averaged two-photon image showing GCaMP6f expression in dorsal CA1 pyramidal neurons in a single animal. Scale bar, 100 μm. Right: Δ*f*/*f* traces from a CA1 place cell (black), and velocity (grey) and licking (orange) signals for five consecutive laps. **b,** Task phases. Left: day 0 is the final habituation day (blank belt with variable reward location). Right: day 1, exposure to a new environment (cue-enriched belt with fixed reward location, that is, environment A). **c,** Licking behaviour of an individual animal. The ticks represent licks; the arrows mark the lap start (left) or lap start and reward location (right). **d,** Mean lick rates for days 0 and 1 (*n* = 18 animals). **e,** Running behaviour of an individual animal. **f,** Mean running for days 0 and 1 (*n* = 18 animals). **g,** Peak-normalized mean Δ*f*/*f* across space for all CA1 place cells (day 0: *n* = 719, day 1: *n* = 1,278). Place cells sorted by peak location. Data from animals with the same field of view imaged in both sessions (*n* = 14 animals).

**h,** Fraction of CA1 place cells (PCs) versus place field (PF) peak location (bin = 18 cm, chi-squared test, df = 9, *P* = 3.47 × 10⁻³⁶). **i,** Fraction of spatially modulated place cells (paired two-sided *t*-test, *P* = 3.12 × 10⁻⁵). **j,** Mean place cell spatial information per animal (paired two-tailed *t*-test, *P* = 0.003). **k,** Population vector correlations (corr.). Left: reliability of CA1 place cell activity (paired two-tailed *t*-test, *P* = 3.22 × 10⁻⁶). Right: population vector correlations for CA1 cells with place fields on days 0 and 1 (two-tailed *t*-test, *P* = 3.65 × 10⁻¹⁵ and 3.82 × 10⁻¹¹). For **h**–**k**, *n* = 14 animals each; in **i**–**k**, open circles show individual animals, and filled circles are means. **l,** CA1 place cell density on day 1 as a function of the peak-normalized velocity (*n* = 18 animals) and fitted by a linear equation (blue line, *R* represents Pearson's correlation coefficient, two-tailed *t*-test, *P* = 4.16 × 10⁻¹⁶). Each dot represents data from an 18-cm-wide spatial bin. Black dashed lines and arrows mark the reward location. Data are shown as mean ± s.e.m. The schematic in **a** has been modified from ref. [17].

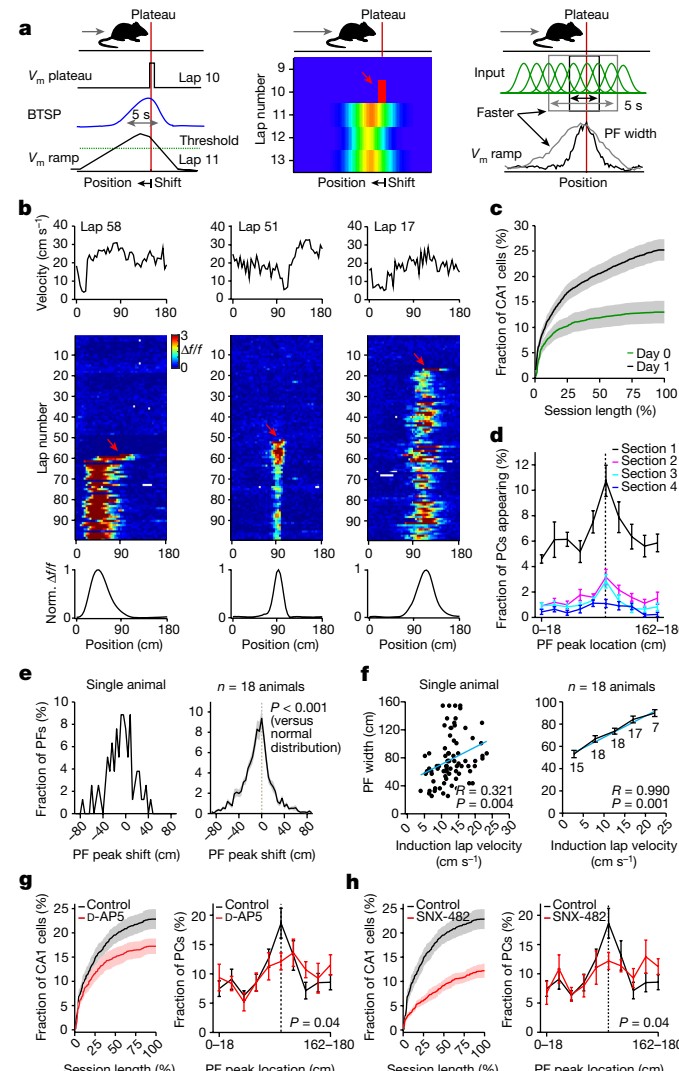

**Fig. 2 | BTSP underlies experience-dependent shaping of CA1 representations. a**, Schematics showing features of BTSP. BTSP's asymmetric time course shifts $V_m$ ramp (left) and place field firing back in space (middle). The faster the mouse runs, the more input will be affected by BTSP and the wider the resulting place field (right). Red lines are the plateau position, grey arrows at the top indicate the direction of the mouse running, and the red arrow indicates the plateau. **b**, Three abruptly appearing place cells. Top: velocity profile of the first lap with place field activity ('induction lap'). Middle: $\Delta f/f$ across laps. The red arrow marks the induction lap. Bottom: peak-normalized (norm.) mean $\Delta f/f$ across space. **c**, Time course of CA1 place cell appearance for days 0 (green) and 1 (black). **d**, Fraction of CA1 place cells as a function of place field peak location and session length ($n = 15$ animals; session divided into four sections of 14–35 laps). **e**, Histograms showing place field peak shift (peak location (cm) of the generated place field minus peak activity location (cm) in the induction lap). Left: individual animal. Right: $n = 18$ animals ($n = 1,727$ CA1 place cells, two-tailed one-sample Kolmogorov–Smirnov test, $P = 1.13 \times 10^{-7}$). **f**, Place field width as a function of the animal's mean induction lap velocity. Left: individual animal. Each dot represents one place cell. Right: $n = 18$ animals. Data are binned into 5 cm s$^{-1}$ velocity bins and fitted by a linear equation (blue line). The $R$ value indicates Pearson's correlation coefficient (left: two-tailed $t$-test, $P = 0.004$; right: two-tailed $t$-test, $P = 0.001$). The numbers indicate the number of data points in each bin. **g**, Effect of an $N$-methyl-D-aspartate receptor antagonist, D-AP5 (50 or 75 μM), on the development of the CA1 representations. Black: control ($n = 10$ animals). Red: D-AP5 ($n = 8$ animals). Left: time course of CA1 place cell appearance. Right: fraction of CA1 place cells as a function of place field peak location (chi-squared test, df = 9; $P = 0.04$). **h**, Effect of a Ca$^{2+}$ channel antagonist, SNX-482 (10 μM). Black: control ($n = 10$ animals). Red: SNX-482 ($n = 7$ animals). The panels show the same as **g** (chi-squared test, df = 9, $P = 0.04$). Black dashed lines depict the reward location. Data are shown as mean ± s.e.m.

of this new type of synaptic plasticity, we observed that previously silent neurons acquired place fields abruptly in a single lap (mean laps before onset: 22.6 ± 0.7; mean laps with activity before onset at field location: 1.2 ± 0.07; Fig. 2b), with a substantial fraction of new place fields added during the learning session (Fig. 2c,d and Extended Data Fig. 1). These suddenly appearing place fields exhibited additional hallmark features of BTSP[12,19]. First, place fields tended to shift backwards in space compared to the activity in their induction lap (Fig. 2e). Second, we observed a linear relationship between the place field's width and the animal's velocity in the induction trial (Fig. 2f). Finally, the development of the experience-dependent representation during the session, including the reward over-representation, was significantly inhibited by local application of a pharmacological antagonist of synaptic plasticity, D-2-amino-5-phosphonovalerate (D-AP5) (Fig. 2g), or an inhibitor of plateau firing, the CaV2.3 channel blocker SNX-482 (Fig. 2h). Presumably, the local nature of the antagonist application limited the behavioural impact of the manipulation as all behavioural measures were unaltered (Extended Data Figs. 2 and 3). The above results indicate that the experience-dependent shaping of the CA1 representation requires BTSP.

## Necessity of EC3

EC3 axons innervate the apical dendritic tuft[20–22], which is the site of plateau initiation in CA1, where they deliver large-amplitude synaptic input that has an elevated ratio of $N$-methyl-D-aspartate receptors to α-amino-3-hydroxy-5-methyl-4-isoxazole propionic acid receptors[23], and previous in vitro and in vivo work has shown that EC3 input regulates both the probability and duration of plateau potentials[14,16]. Therefore, we next questioned whether perturbing EC3 input could affect the formation of the CA1 over-representation. To examine this, we used a retrograde virus infection strategy[24] to express the hyperpolarizing proton pump archaerhodopsin-T (ArchT)[25] only in the subset of EC3 neurons that projected to our recording area in CA1 (Fig. 3a and Extended Data Fig. 4). The intent of this strategy was to minimize the impact on overall entorhinal cortex activity and mouse behaviour by affecting only a prescribed group of EC3 neurons. As a control, we expressed tdTomato alone instead of ArchT–tdTomato in a separate group of mice that, otherwise, received the same treatment. We found that inhibiting this subset of EC3 axons by delivering 594-nm laser light (40 Hz, sinusoidal stimulation)[26] to the entorhinal cortex in a zone of 36 cm (±18 cm) around the reward prevented the development of the CA1 reward over-representation compared to the control group (Fig. 3b,c and Extended Data Fig. 5). Notably, there was no significant change in the amplitude of place fields near the reward zone ($n = 6$ mice (tdTomato) versus $n = 8$ mice (ArchT), 78% versus 84% $\Delta f/f$, two-tailed unpaired $t$-test, $P = 0.401$), mean Ca$^{2+}$ event amplitude (Extended Data Fig. 5c, $n = 6$ mice (tdTomato) versus $n = 8$ mice (ArchT), two-tailed unpaired $t$-test, $P = 0.06$), the time course of place field formation (Extended Data Fig. 5d) or in the licking and running behaviours (Extended Data Fig. 5e) between the control mice and the ArchT group.

## EC3 activity pattern

As EC3 seems to be necessary for the experience-dependent shaping of the CA1 representation, we next examined the activity of EC3 neurons projecting to CA1. The axons of these cells are located in the stratum lacunosum-moleculare of dorsal CA1 (Fig. 3d and Extended Data Fig. 4) and are accessible for two-photon imaging through our standard hippocampal window. Thus, we performed axonal two-photon Ca$^{2+}$ imaging in mice expressing GCaMP6f in EC3 neurons (Fig. 3e and Extended Data Fig. 6). A moderate level of selective activity was observed in individual axons as spatial (average $\Delta f/f$ maximum/mean) and velocity (Pearson's correlation coefficient between average $\Delta f/f$ and average velocity) tuning[27–30] (Fig. 3f–i). However, when the average axon $\Delta f/f$

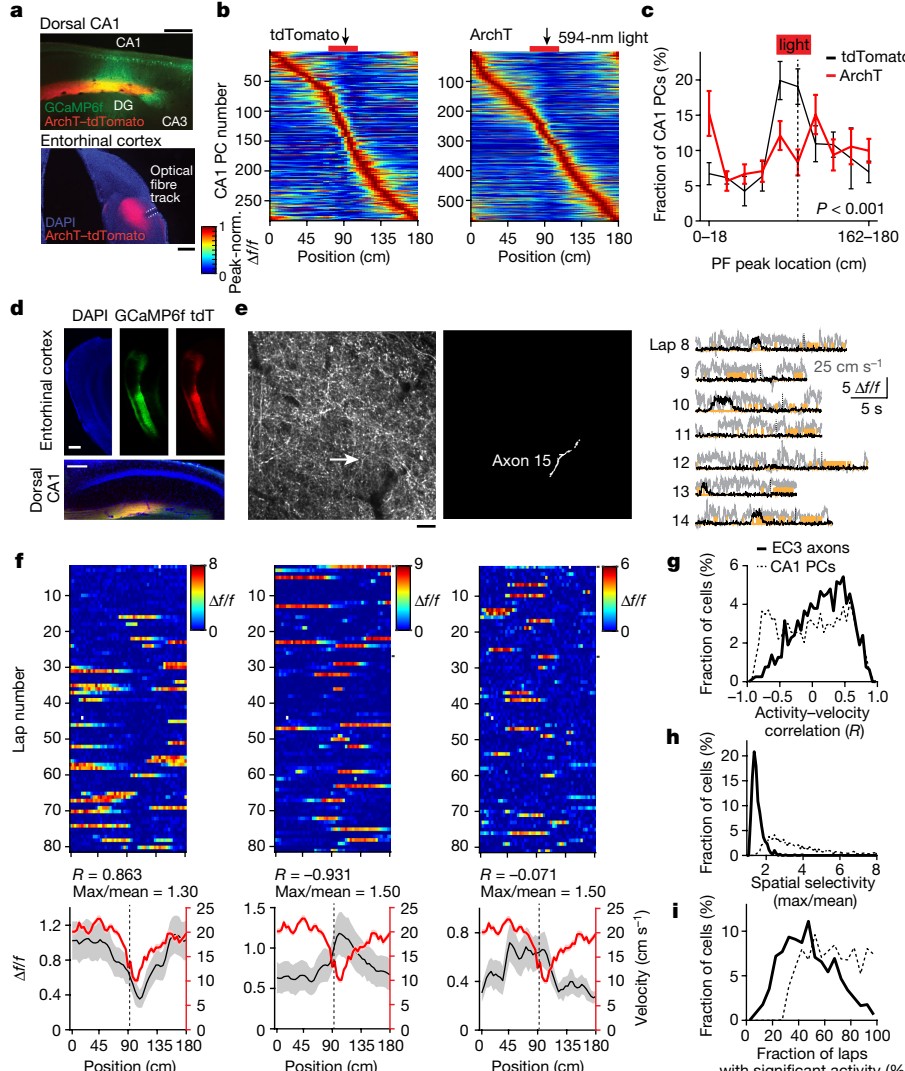

**Fig. 3 | EC3 activity is required for the experience-dependent shaping of CA1 representations. a–c**, Ipsilateral optogenetic perturbation of EC3 neuronal activity prohibits the development of the reward over-representation. Black: tdTomato control ($n = 6$ animals); red: ArchT ($n = 8$ animals). **a**, Viral expression of GCaMP6f in CA1 (top) and ArchT–tdTomato in EC3 (bottom). EC3 axons can be found in the stratum lacunosum-moleculare of CA1 (top). DG, dentate gyrus; DAPI, 4′,6-diamidino-2-phenylindole. Scale bars, 250 μm (top) and 500 μm (bottom). **b**, Peak-normalized mean $\Delta f/f$ across space for all CA1 place cells. The red bar marks light-on locations. **c**, Fraction of CA1 place cells as a function of place field peak location (chi-squared test, df = 9, $P = 8.82 \times 10^{-19}$). **d–i**, Recording of EC3 axonal activity in the stratum lacunosum-moleculare of CA1. **d**, Representative images showing viral expression of GCaMP6f and tdTomato (tdT) in EC3 neurons (top) and their axons in hippocampal area CA1 (bottom) in $n = 1$ animal. Scale bars, 350 μm (top) and 200 μm (bottom) **e**, Left: representative two-photon, time-averaged image showing expression of GCaMP6f in EC3 axons in a single animal. Scale bar, 20 μm. White arrow depicts the location of axon 15. Middle: white area depicting axon 15. Right: Ca²⁺ $\Delta f/f$ traces (black) for seven consecutive laps recorded from axon 15. Simultaneously recorded velocity and licking signals are shown in grey and orange. **f**, Three individual EC3 axons. Top: $\Delta f/f$ across laps. Bottom: Mean $\Delta f/f$ (black) and mean velocity (red) across space. Black and red $y$ axes apply, respectively. EC3 axons are classified on the basis of their mean $\Delta f/f$–mean velocity correlation (Pearson's correlation coefficient, $R$) and spatial selectivity index (maximum (max) divided by the mean of the mean $\Delta f/f$ trace). **g–i**, Distributions of the activity–velocity correlations (**g**), the spatial selectivity indices (**h**) and the fractions of laps with significant activity (EC3: all locations included, CA1 place cells: only place field locations included) (**i**) from 792 axons from 7 animals (solid line) and 1,727 CA1 place cells from 18 animals (dashed line). Black arrows and dashed lines depict the reward location. Data are shown as mean ± s.e.m.

was compared for interleaving trials (median Pearson's correlation coefficient = 0.247, odd versus even trials, Fig. 4a,b), only a fraction of axons (19%) showed relatively well-correlated firing (peak locations on even trials were within 10 cm of odd trials; Extended Data Fig. 7a,b). The peak firing locations of both the general population of EC3 axons and the well-correlated axons uniformly tiled the entire track (Fig. 4a–c and Extended Data Fig. 7a–c). Similar results were also observed for the 5% most selective EC3 axons[28] (Extended Data Fig. 7d–f). Finally, in contrast to the above uniform distributions, the spatial selectivity and axon–axon correlation values (odd versus even laps) of EC3 axons

were significantly elevated around the reward locations (Fig. 4d, reward at 90 cm). These data indicate that the activity of most EC3 neurons during this behaviour showed a moderate level of tuning along with a substantial degree of stochasticity and that, although the spatial distribution of these tuned neurons was uniform across the environment, the level of spatial tuning was elevated around the reward site (Fig. 4a–d).

To understand how this EC3 activity pattern could affect postsynaptic CA1 pyramidal neurons, we turned to computational modelling (Extended Data Fig. 7g–j). As EC3 single-axon activity was indicative of a stochastic process (for example, exponentially distributed activity times

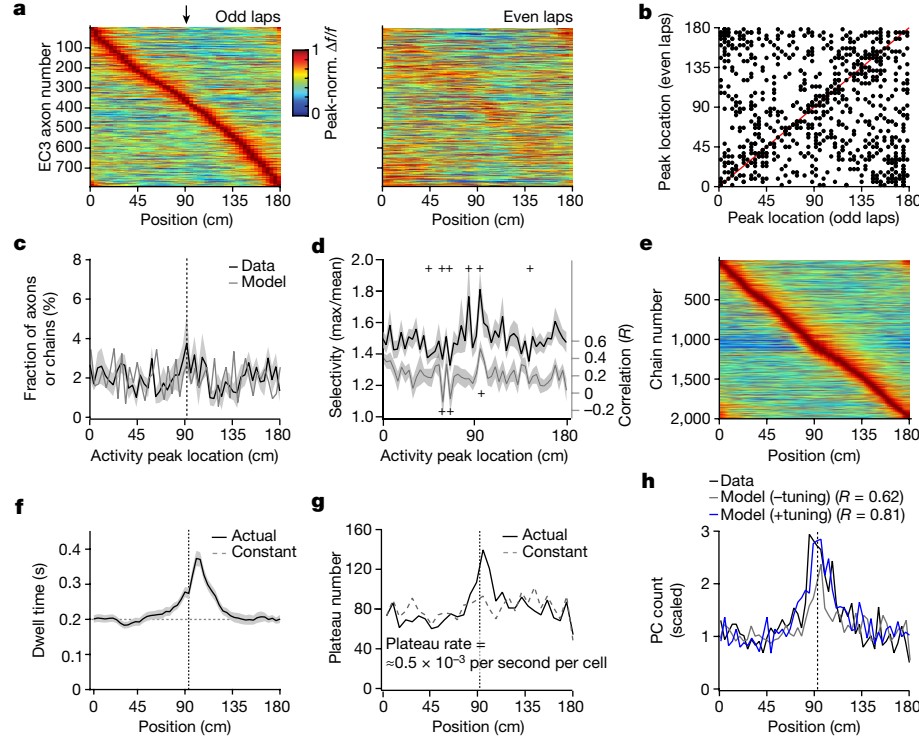

**Fig. 4 | Elements of EC3 activity shaping CA1 representation.**
**a**, Peak-normalized mean $\Delta f/f$ across space for all EC3 axons recorded ($n = 792$ axons in 7 animals). Heat maps for odd and even laps are shown separately. EC3 axons are ordered according to their peak location during the odd laps.
**b**, Scatter plot showing axonal activity peak locations for averages made from odd and even laps. The unity line is depicted in red. **c**, Fraction of EC3 axons and chains as a function of activity peak location (data: black, Markov model: grey). The track is divided into 50 spatial bins of 3.6 cm each. **d**, EC3 axon tuning (black) and odd–even lap correlation per axon (grey) across space. The plus symbols indicate values that lie outside the 95% confidence interval generated from 1,000 data shuffles of $n = 792$ data points. **e**, Sorted peak-normalized

mean activity for 2,000 modelled Markov chains. **f**, Mean dwell time profile for $n = 18$ animals (black); the dashed grey line is the dwell time for a constant running speed. **g**, The model predicts the number of threshold crossings (dendritic plateau potentials) observed across space. Black solid: experimentally observed running profile. Grey dashed: constant running speed. **h**, Scaled CA1 place cell count across space. Black: data. Grey: model without enhanced tuning added. Blue: model with enhanced tuning. Pearson's correlation coefficients ($R$) between the data and model are indicated. Black arrows and dashed lines depict the reward location. Data are shown as mean ± s.e.m.

and low cell–cell correlations, Fig. 4a,b and Extended Data Fig. 7a–c,j), we modelled the activity of individual EC3 axons as simple two-state Markov chains (Extended Data Fig. 7g–j). This approach simulates EC3 neuron activity transitions between an inactive and active state, similar to the persistent firing previously observed in this region[30–33]. To produce the distributed set of cell–cell correlations, we adjusted the activation transition probability uniformly across the track ($P_{01}$ increased from 0.04 to 0.20 for ten time steps at a point in the lap that incremented smoothly across the population; Fig. 4c,e and Extended Data Figs. 7k–p and 8a–h).

We next used the model to examine the influence of each of the observed elements of the EC3 activity on the probability that a given CA1 postsynaptic neuron receives a suprathreshold, plateau-evoking amount of EC3 input (Extended Data Fig. 8i–k). When using a steady running speed (Fig. 4f; about 0.2 s dwell time per spatial bin), the simulation predicts that the uniform level of EC3 input across the track produces a steady probability of plateau potential initiation (Fig. 4g). Hence, when using the animals' actual spatial running profile (Fig. 4f, solid line; total lap time equalled 10 s in the model), the fraction of neurons initiating plateaus at the reward site increases approximately twofold, simply because the animals spend approximately twice the time at this location (Fig. 4g). Although the spatial distribution of CA1 place cells predicted by the model was highly correlated with the actual observed distribution (Fig. 4h), the increased density of CA1 place cells at the reward site was higher in the data than predicted by the model (about threefold versus about twofold). We, therefore, next included the observed enhanced

level of EC3 neuronal tuning and stability around the reward site ($P_{01}$ was elevated from 0.20 to 0.68 in 100 chains with peak firing near the reward site; Fig. 4d and Extended Data Fig. 8h) and found that the CA1 place cell distribution now predicted by the model was even more accurate[34,35] (Fig. 4h). We conclude that three primary elements related to EC3 activity shape learning-related changes in CA1 population activity. These are a spatially uniform distribution of moderately tuned EC3 neuronal activity, a nonuniform level of spatial tuning in these same EC3 neurons (enhanced tuning around the reward site) and the nonuniform running behaviour of the animal (increased dwell time around the reward site).

## Environmental dependence of EC3 activity

Notably, the first of the above elements, the spatial distribution of EC3 activity, reflects the uniformity of sensory cues in the environment. To determine whether the EC3 activity driving CA1 plasticity is related to the spatial profile of sensory cues in the environment, we examined how EC3 activity responds to a less uniform environment containing only a single prominent, new and reward-predictive feature. We, thus, designed a different environment (environment B) that included a visual stimulus (10-Hz blue light flashes for 500 ms to both eyes) activated 50 cm before the fixed reward delivery site and no other experimenter-placed belt cues (Fig. 5a, bottom). This second environment elicited subtle alterations in the licking of the mice (Fig. 5b) and more substantial changes in their running (Fig. 5c). In addition, the EC3 axon population activity was heavily altered. The most prominent

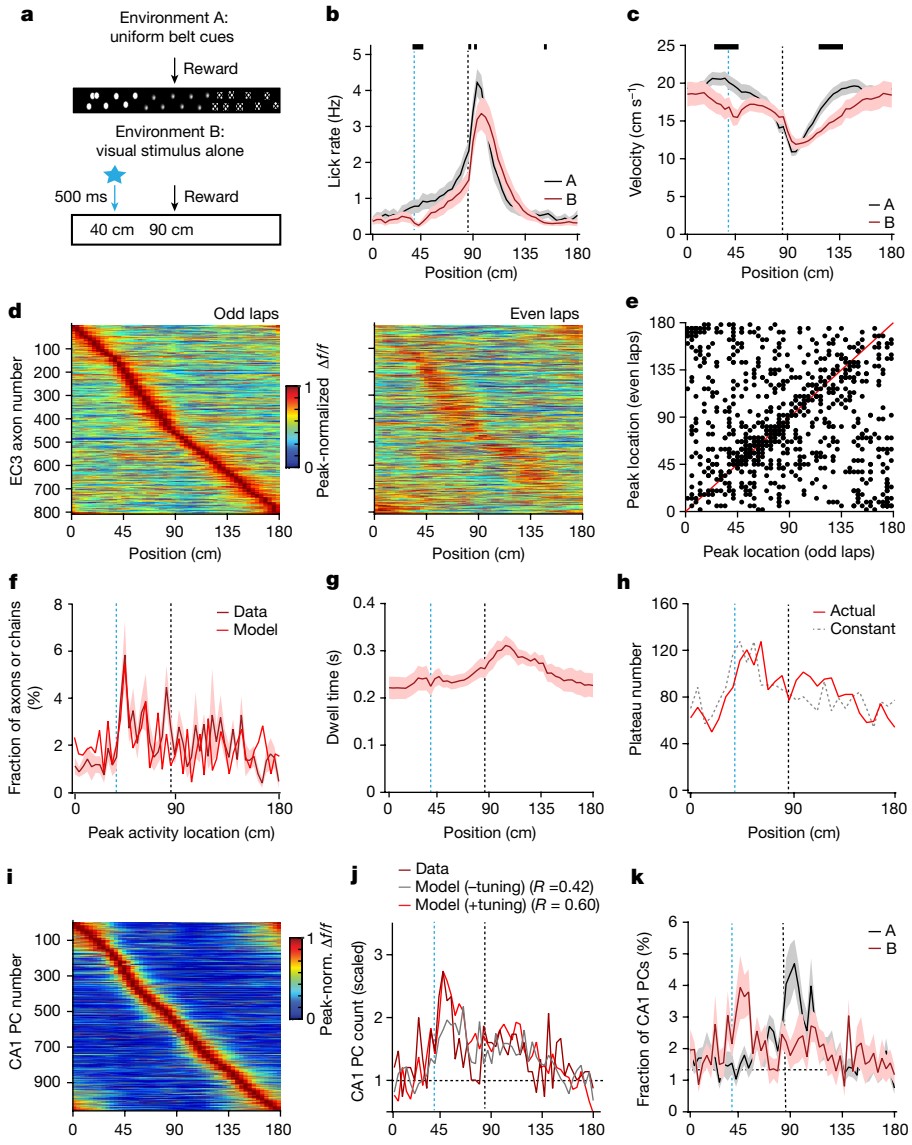

**Fig. 5 | EC3 activity adapts to the environment. a**, In contrast to environment A (top), environment B (bottom) involves a blank belt and a visual stimulus (blue light-emitting diode flashes, 10 Hz, 500 ms) 50 cm before the single, fixed, reward (black arrow). **b**, Mean lick rate in environment A (black, $n = 25$ mice) and B (maroon, $n = 17$ mice). **c**, Mean running profile in environments A (black) and B (maroon). **d**, Peak-normalized mean $\Delta f/f$ across space for EC3 axons ($n = 808$, $n = 8$ animals) in environment B. Colour plots for odd and even laps are shown separately. EC3 axons are ordered according to their peak location during the odd laps. **e**, Scatter plot showing axonal activity peak locations for averages made from odd and even laps. The unity line is depicted in red. **f**, Histogram of the peak activity locations of all EC3 axons (maroon, solid) and modelled Markov chains (red). The track is divided into 50 spatial bins of 3.6 cm each. **g**, Mean dwell time profile for $n = 9$ animals. **h**, The model

predicts the number of threshold crossings (dendritic plateaus) observed (red: experimentally observed running profile; grey: constant velocity). **i**, Peak-normalized mean $\Delta f/f$ across space for all CA1 place cells ($n = 1,058$, $n = 9$ animals) in environment B. **j**, Scaled CA1 place cell count. Maroon: data. Grey: model without enhanced tuning added. Red: model with enhanced tuned. Pearson's correlation coefficients ($R$) between the data and model are indicated. **k**, Fraction of CA1 place cells as a function of place field peak location (A: $n = 18$, black; B: $n = 9$, maroon; chi-squared test, df = 49, $P = 3.81 \times 10^{-33}$). Black bars in **b**,**c** indicate locations with $P < 0.05$ (unpaired two-tailed $t$-test, performed on each spatial bin). Blue dashed lines depict the light onset; black dashed lines depict the reward location. Data are shown as mean ± s.e.m.

change was an approximately fourfold increase in the fraction of axons whose firing peaked around the visual stimulus (Fig. 5d–f and Extended Data Fig. 9a–e). Nevertheless, EC3 axon activity retained a high degree of instability, with only a fraction of axons (25%) showing relatively consistent firing between interleaved trials (Fig. 5d,e and Extended Data Fig. 9c,d). The density of these well-correlated axons was enriched at the location of the light stimulus (Fig. 5e and Extended Data Fig. 9c,d), as were both the level of spatial tuning and activity stability (Extended Data Fig. 9f). These data indicate that the activity of the EC3 neurons reflected the distribution of the relevant environmental cues.

To capture this new EC3 activity data more accurately, we adjusted the previous Markov chain simulation by increasing the fraction of chains that had an elevated activation probability around the visual cue position such that the final spatial profile of peak chain activity showed an approximately fourfold increase around this position (Fig. 5f, Methods and Extended Data Fig. 9g–n). Moreover, this modification recapitulated the increased density of the well-correlated axons around the visual stimulation location (Extended Data Fig. 9l). When we ran the simulation using this nonuniform density of EC3 activity and the actual running behaviour of the mice in environment B (Fig. 5g; total

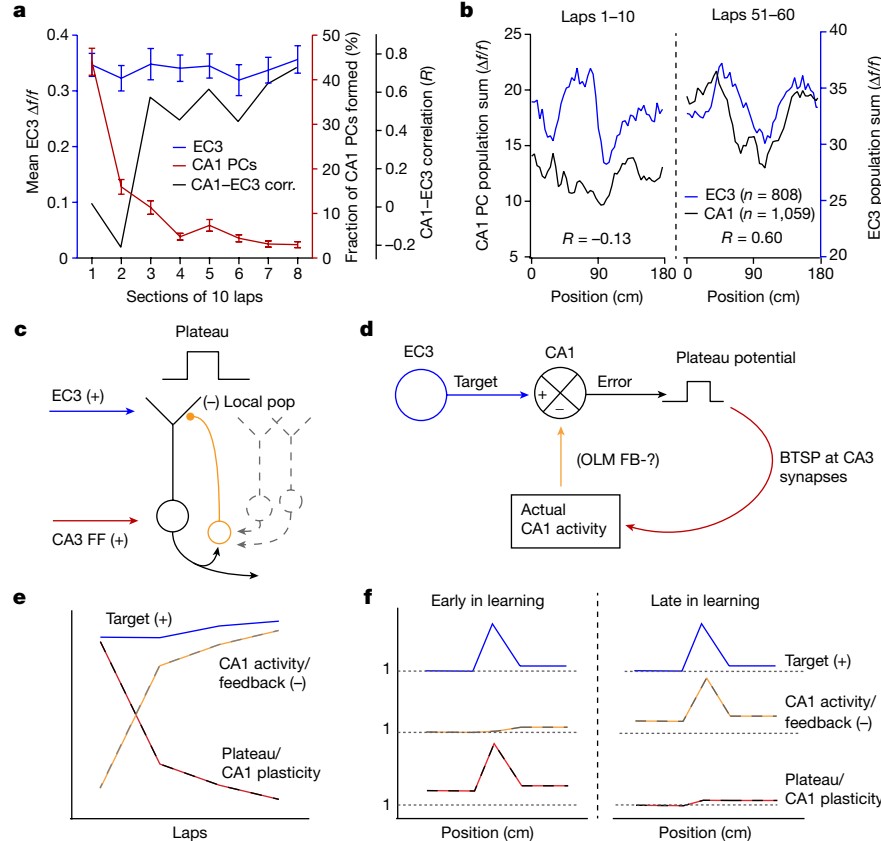

**Fig. 6 | EC3 provides a target signal for CA1 population activity. a**, Mean EC3 $\Delta f/f$ around the over-representation (blue), the fraction of CA1 place cells formed (maroon) and the correlation between summed EC3 and CA1 population activity (black) for the first eight session sections (10 laps each). Data from $n = 27$ mice (CA1 place cells) and $n = 15$ mice (EC3 axons) were included. **b**, Summed EC3 (blue, $n = 808$ axons) and CA1 (black, $n = 1,059$ CA1 place cells) population activity in environment B for laps 1–10 (left) and 51–60 (right). **c**–**f**, Proposed network scheme for learning in CA1. **c**, CA1 circuit elements involved include CA1 pyramidal neurons (black solid and grey dashed), excitatory input from EC3 innervating the distal apical tuft regions (blue), excitatory feedforward (FF) input from CA3 whose synaptic weights are adjusted during learning innervating the perisomatic regions (red), oriens lacunosum-moleculare feedback inhibitory interneurons bringing a copy of regional population (pop.) CA1 activity to the apical tuft (orange). **d**, EC3

provides each individual CA1 neuron with a desired or target activity pattern that is compared in the distal apical dendrites with a representation of the actual pattern of population activity that could be provided by oriens lacunosum-moleculare (OLM) feedback (FB) interneurons. An excess excitation (mismatch) will increase the probability of dendritic $Ca^{2+}$ plateau potential initiation. The plateau functions as a local error signal in each CA1 cell driving BTSP at CA3 feedforward excitatory inputs (learning pathway) to shape the firing of each CA1 cell accordingly. The altered CA1 population activity feeds back to the comparator in the apical tuft. **e**, Temporal profiles of proposed signals. The EC3 target signal (blue) remains stable, whereas the plateau probability driving CA1 plasticity (black–red) decreases, as the CA1 representation and the inhibitory feedback signal (orange–grey) increases. **f**, Spatial profiles of proposed signals. Unless indicated otherwise, data are shown as mean ± s.e.m.

lap time equalled 10 s in the model) to infer plateau probability across space, our results predicted the presence of an over-representation of the visual stimulus location that was even larger than that at the reward site (Fig. 5h).

We tested this prediction by performing $Ca^{2+}$ imaging in CA1 pyramidal cells in mice exposed to environment B. We first found that the basic characteristics of place cell activity, such as the fraction of spatially modulated cells and the place cell spatial information content per animal, remained unchanged (Extended Data Fig. 10). However, consistent with our modelling data, our observations showed that the largest fraction of place cells was near the location of the light cue (Fig. 5i–k) and the spatial place cell distribution was significantly different from that observed in environment A (Fig. 5k). A model containing each of the three primary elements of EC3 activity (the observed increased density and enhanced neuronal tuning around the predictive cue location as well as the actual running profile of the animals) produced the most accurate prediction of place field distribution (Fig. 5j). Thus, it seems that the presence of a unique predictive cue in environment B heavily altered the spatial pattern of EC3 neuronal activity and CA1 plasticity responded accordingly to produce a unique population activity profile.

The necessity of EC3 input was confirmed by optogenetically inhibiting EC3 activity around the predictive visual cue, which eliminated the CA1 over-representation of the light cue (Extended Data Fig. 11). These results corroborate our hypothesis that EC3 is necessary for shaping the CA1 place field representation. Furthermore, it seems that the spatial tuning of EC3 neuronal activity is sensitive to behaviourally relevant aspects of an environment[28,36].

## Form of EC3 instructive signal

The above data suggest that EC3 input directs neuronal plasticity in CA1 by providing a type of instructive signal. Therefore, we next attempted to determine the form of this EC3 instructive signal. If it is functioning as an error signal, we would expect EC3 activity around the over-representation site to decrease as CA1 population activity approached the desired pattern. On the other hand, if the EC3 provides a signal representing the desired CA1 activity pattern (a target signal) to each CA1 pyramidal neuron, it should remain more constant throughout the session, even as CA1 plasticity decreases[37]. To examine this, we plotted EC3 population activity (Fig. 6a, blue) as a function

of the session duration alongside the plasticity of CA1 place cells (Fig. 6a, maroon). We found that although the formation of new CA1 place cells decreased markedly during the session, the EC3 activity profile remained steady throughout. In addition, the overall CA1 activity pattern rapidly approached that of the constant EC3 activity profile (Fig. 6a, black, and Fig. 6b). Together, these results indicate that the EC3 provides a relatively invariant instructive signal that is more reminiscent of a target signal than an error signal. In our current scheme, the excitatory target from EC3 combines in the apical dendrites with an inhibitory signal representing the actual CA1 population activity, and the resulting plateau potentials function as a local error signal that is unique in each CA1 neuron[13,37,38] (Fig. 6c,d). The source of the actual CA1 activity signal remains undetermined and may involve inhibitory, neuromodulatory, disinhibitory or other elements[5,22,39–44]. The proposed evolution of these signals during a learning session is shown in Fig. 6e,f. We conclude that EC3 provides a target signal that instructs CA1 in how to represent the environment during a spatial learning task.

## Summary and conclusions

This work addresses the long-standing question of what neural mechanisms underlie learning within the mammalian brain. In the past, we have observed that optogenetic inhibition of EC3 input reduces plateau potential initiation in CA1 cells[16]. There is considerable data, presented here and previously, showing that plateau potentials induce BTSP and place field formation[7,12,13,16–19,45,46]. Finally, we now report that inhibition of EC3 input reduces place field formation and alters experience-dependent shaping of CA1 representations. Given all of the evidence, we conclude that EC3 activity drives plateau potentials in CA1 neurons to induce new place field formation through BTSP and that this is the primary mechanism by which learning-related changes in CA1 population activity occur. Several lines of evidence presented above suggest that EC3 functions as a target-like instructive signal that directs BTSP to achieve a particular desired CA1 population activity. Notably, this EC3 target-like activity reflected the distribution of salient environmental cues, which ranged in uniformity. Further experiments are required to determine exactly how EC3 neurons are able to produce an environmentally specific instructive signal.

Target signals can theoretically be quite powerful in directing learning in complex neuronal networks because they provide a means to account for the multitude of downstream parameters that lie between regional activity and desired behaviour[47,48]. However, reports of target signals driving synaptic plasticity, Hebbian or otherwise, are rare[37]. Indeed, even brain regions thought to use supervised motor learning have been found to use error signals, not targets[49,50]. The observation that synaptic plasticity directed by adapting target signals shapes the activity of the mammalian hippocampus, an area well known for its importance in spatial learning and episodic memory, raises the possibility that many brain regions may learn in a manner substantially different from that thought at present.

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

## Methods

All experiments were carried out according to methods approved by the Institutional Animal Care and Use Committees at Janelia (protocols 12–84 and 15–126) and the Baylor College of Medicine (protocol AN-7734).

### Surgery

All experiments were carried out in adult (older than postnatal day 66 at the time of surgery) GP5.17 (ref. [11]; $n = 52$ mice, Janelia and Jackson Laboratories) or pOxr1–Cre (ref. [22]; $n = 44$ mice, Jackson Laboratories) mice of either sex by an experimenter who was not blind to the experimental conditions. Animals were housed under an inverse 12-h dark/12-h light cycle (lights off at 9 am) in the Magee laboratory satellite facility with the temperature (about 21 °C) and humidity (about 30–60%) controlled. All surgical procedures were performed under deep isoflurane anaesthesia. After locally applying topical anaesthetics, the scalp was removed, and the skull was cleaned. Then the skull was levelled, and the locations for the craniotomies were marked using the following stereotactic coordinates: 1) centre of the 3-mm-diameter hippocampal window: 2.0 mm posterior from bregma and 2.0 mm lateral from the midline; 2) CA1 virus injections: 2.0 mm posterior from bregma and 1.9 mm lateral from the midline and 2.3 mm posterior from bregma and 2.2 mm lateral from the midline; 3) entorhinal cortex virus injections: 4.7 mm posterior from bregma and 3.5 mm from the midline and 4.7 mm posterior from bregma and 3.8 mm from the midline; 4) entorhinal cortex optical fibre implantation: 4.7 mm posterior from bregma and 4.4 mm from the midline; and 5) entorhinal cortex local field potential (LFP) recordings: 4.7 mm posterior from bregma and 3.5 mm from the midline. Then, for all experiments except the LFP recordings, a 3-mm-diameter craniotomy was made above the hippocampus. Cortical tissue within the craniotomy was slowly aspirated under repeated irrigation with warmed sterile saline 0.9%. Once the external capsule was exposed, the cannula (3 mm diameter, 1.7 mm height) with a window (CS-3R, Warner Instruments) on the bottom was inserted and cemented to the skull. Finally, a custom-made titanium head bar was attached to the skull using dental acrylic (Ortho-Jet, Lang Dental).

For the experiments with GCaMP6f and tdTomato expression in EC3 or ArchT or tdTomato expression in EC3 and GCaMP6f expression in CA1, the hippocampal window surgery was preceded in the pOxr1–Cre mice[22] ($n = 44$) by ipsilateral virus injections using the coordinates stated above. Notably, the pOxr1–Cre mouse line expresses Cre recombinase predominantly in the medial entorhinal cortex. For the virus injections, we first made a small (about 0.5-mm diameter) craniotomy. This was followed by injecting a small volume of one of the following mixtures (all viruses produced by the Janelia Viral Vector Core; viral titres range between 1 and $7.5 \times 10^{12}$): AAV1.Syn.GCaMP6f.WPRE.SV40 and AAVrg.Syn.Flex.ArchT–tdTomato.WPRE.SV40 into area CA1 (dorsoventral: 1,350 and 1,000 µm; 25 nl per depth); AAV1.Syn.GCaMP6f.WPRE.SV40 and AAVrg.Syn.Flex.tdTomato.WPRE.SV40 into area CA1 (dorsoventral: 1,350 and 1,000 µm; 25 nl per depth); AAV1.Syn.Flex.GCaMP6f.WPRE.SV40 and AAV1.Syn.Flex.tdTomato.T2A.tdTomato.WPRE.SV40 into the entorhinal cortex (dorsoventral: 2,100, 1,800 and 1,500 µm; 50 nl per depth). All injections were followed by a waiting period of 6 min about 300 µm above the last injection depth. The injection system comprised a pulled glass pipette (broken and bevelled to 15–20 µm (outside diameter); Drummond, Wiretrol II Capillary Microdispenser), backfilled with mineral oil (Sigma). A fitted plunger was inserted into the pipette and advanced to displace the contents using a manipulator (Drummond, Nanoject II). Retraction of the plunger was used to load the pipette with the virus. The injection pipette was positioned with a Sutter MP-285 manipulator. For the optogenetics experiments, an optical fibre (core diameter of 200 µm) was chronically implanted at a 45° angle into the ipsilateral entorhinal cortex (at a depth of 50–100 µm) and attached to the skull using dental cement (Calibra Dual Cure, Pearson Dental).

### Behavioural training and task on the linear track treadmill

The linear track treadmill consisted of a belt made from velvet fabric (McMaster Carr). The belt (length of 180 cm) was self-propelled, and the reward was delivered through a custom-made lick port controlled by a solenoid valve (Parker). The animal's speed was measured using an encoder attached to one of the wheel axles. A microprocessor-based (Arduino) behavioural control system interfaced with a MATLAB graphical user interface controlled the valve, the sensors and the encoder. In addition, a separate microprocessor (Arduino) interfaced with a MATLAB graphical user interface was used to operate the laser shutter for the optogenetic perturbation experiments and control the visual stimulation according to the animal's position on the belt. Behavioural data were monitored and recorded using a PCIe-6343, X series data acquisition system (National Instruments), and the Wavesurfer software (version 0.982, Janelia).

At 5–7 days after the optical window implantation, running wheels were added to the home cages, and mice were placed on water restriction (1.5 ml per day). After both training and recording sessions, mice were supplemented with additional water to guarantee a 1.5 ml per day water intake. After 5–6 days of familiarizing the animals with the experimenter, mice were trained to run head-fixed on the linear treadmill for 3–5 days. This training was conducted during the animals' dark cycle, and mice were trained on a blank belt (no sensory cues) to run for a 10% sucrose solution reward delivered at lap-to-lap varying locations.

To record neuronal activity and study the development of CA1 representations as mice learned to navigate in a new environment, we exposed the animals to two different environments ('day 1'). Environment A consisted of a belt enriched with three different visual and tactile cues (glue sticks, Velcro tape patches and white dots), which covered the entire length of the belt[12,16,17]. The reward was delivered at a single, fixed reward location. For environment B, the belt was devoid of any local cues, and a bilateral visual stimulus (blue light-emitting diode positioned in front of both eyes, flashing at 10 Hz for 500 ms) was delivered 50 cm before the fixed reward location. Individual recording sessions lasted between 45 and 60 min, with one recording session per day.

### In vivo two-photon Ca²⁺ imaging

All $Ca^{2+}$ imaging recordings were performed in the dark using a custom-made two-photon microscope (Janelia MIMMS design). GCaMP6f and, if expressed, tdTomato were excited at 920 nm (typically 40–70 mW) by a Ti:sapphire laser (Chameleon Ultra II, Coherent) and imaged through a Nikon 16×, 0.8-numerical-aperture objective. Emission light passed through a 565 DCXR dichroic filter (Chroma) and either a 531/46 nm (GCaMP channel, Semrock) or a 612/69 nm (tdTomato channel, Semrock) bandpass filter. It was detected by two GaAsP photomultiplier tubes (11706P-40SEL, Hamamatsu). Images ($512 \times 512$ pixels) were acquired at about 30 Hz using the ScanImage software (R2015 and R2018, Vidrio).

For CA1 pyramidal neuron $Ca^{2+}$ imaging, imaging fields (size varied from $280 \times 280$ to $380 \times 380$ µm) were selected on the basis of the presence of $Ca^{2+}$ transients in the somata. One field of view was imaged per day. If possible, the same field of view was imaged on days 0 and 1 ($n = 14/18$ animals).

For EC3 axonal $Ca^{2+}$ imaging, imaging fields (size of $230 \times 230$ µm) were selected on the basis of the presence of the fibre morphology in the tdTomato channel and the occasional $Ca^{2+}$ transient in the field of view. No attempt was made to locate the same imaging field from day to day.

### Local pharmacology during two-photon imaging

For the local pharmacology experiments, the animal was briefly anaesthetized about 45 min before the recording session using isoflurane.

Then the hippocampal window was carefully punctured (about 50–100-μm-wide hole) near the imaging field of view. This procedure lasted about 5–10 min. In the case of the D-AP5 experiments, the hole was then covered with a silicone elastomer (Kwik-Cast, wpi), and the animal was allowed to recover from the anaesthesia for about 45 min. Then, after positioning the animal under the two-photon microscope, we removed the Kwik-Cast plug and filled the cannula either with D-AP5 (50 or 75 μM) dissolved in sterile saline or with sterile saline alone. The animal was prevented from running for the initial 5–10 min to allow for the initial diffusion of the drug. D-AP5 continued to be present in the cannula throughout the recording session. In the case of the SNX-482 experiments, the hippocampal window was also punctured. We then injected about 50 nl of either SNX-482 (10 μM) dissolved in sterile saline or sterile saline alone onto the distal apical dendritic region of CA1 (injection depth of about 320 μm below the hippocampal surface), using the same procedure as described above for the virus injections. Subsequently, the hole was covered with Kwik-Cast, and the animal was allowed to recover for about 45 min. Two-photon Ca²⁺ imaging then proceeded as usual. Notably, there was no difference in the licking or running behaviours between the standard experiments and those involving local pharmacology (Extended Data Figs. 2 and 3).

## Optogenetic perturbation of EC3 activity

To preferentially manipulate EC3 activity, *loxP*-flanked ArchT[25] driven by a synapsin promoter was virally expressed by injecting AAVrg carrying the *loxP*-flanked ArchT–tdTomato payload into the area CA1 of pOxr1–Cre mice (see above)[22], which express Cre recombinase mostly in layer 3 neurons of the medial entorhinal cortex. The hippocampal window was implanted during the same surgery, and a fibre-containing ferrule was inserted into the entorhinal cortex. The ferrule contained a 200-μm-core, 0.5-numerical-aperture, multimode fibre (FP200ERT, Thorlabs) and was constructed using published techniques[51]. Approximately 21 days after virus injection, combined two-photon imaging and optogenetic experiments were carried out. ArchT was activated using light pulses (maximal duration of 5 s, 594 nm, 40 Hz, sinusoidal pattern, Mambo laser, Cobolt), delivered through the optical fibre located in the entorhinal cortex. The mean laser power was 5–10 mW (ref. [26]; measured each day before the recording in air, about 0.5 cm from the tip of the fibre optic patch cable). As a control, the fluorescent protein tdTomato was virally expressed in pOxr1–Cre mice. These control mice were treated the same as the ArchT group.

To confirm an effect of the ArchT activation on EC3 activity, we carried out LFP recordings in the entorhinal cortex of a group of mice (n = 4) that expressed ArchT in EC3. Glass electrodes (1.5–3.5 MΩ) were filled with 0.9% saline and mounted vertically on a micromanipulator (Luigs & Neumann). The LFP signal was monitored using an audio amplifier (Grass Technologies), while the electrode was advanced slowly through the brain with about 0.5 psi of pressure. The LFP recording locations were about 1.7 mm below the cortical surface. Once this depth was reached, we removed the pressure and started recordings. We alternated between control laps without and laps with ArchT activation. There was no randomization in the sequential ordering of control laps and laps with light application.

## Histology

Mice were transcardially perfused with phosphate-buffered saline (PBS) or Dulbecco's PBS, followed by a 4% paraformaldehyde solution. Extracted brains remained overnight in 4% paraformaldehyde and were then rinsed twice and stored in PBS. Then, 50-μm-thick coronal or sagittal sections of paraformaldehyde-fixed brains were made and mounted on glass slides using Fluoromount mounting medium. All histological images were acquired on the ZEISS Zoom.V16 microscope, equipped with ZEN 3.1 software. Histological sections (Extended Data Fig. 4) were analysed using a stereotaxic mouse brain atlas[52] and ImageJ's line plot function (ImageJ version 2.0.0).

## Data analysis

**Ca²⁺ signal extraction and activity map generation.** To extract somatic Ca²⁺ signals of CA1 pyramidal neurons, videos were motion-corrected using SIMA (version 1.3.2)[53], regions of interest (ROIs) were manually drawn to include single neurons (using ImageJ version 2.0.0), and Ca²⁺ traces across time were extracted again using SIMA (version 1.3.2)[53]. To extract axonal EC3 Ca²⁺ signals, the automatic motion correction and ROI detection algorithms of the Suite2P (version 0.6.16) pipeline[54] were used. The output was manually curated for both recording types, and ROIs with insufficient signal were removed. Only datasets for which the motion correction was successful were included in this study. Further analyses of CA1 and EC3 activity were then performed using custom functions written in MATLAB (version 2019a). These included: 1) conversion to $\Delta f/f$, which was calculated as $(F − F0)/F0$, in which $F0$ is the mode of the histogram of $F$; 2) in the case of the axonal Ca²⁺ data, a noise correlation analysis using a Pearson's correlation coefficient threshold of 0.4–0.5 to identify ROIs that probably originate from the same axon/neuron (step-by-step procedure illustrated in Extended Data Fig. 6a–d); Ca²⁺ signals from ROIs belonging to a single axon were combined, and an average Ca²⁺ signal per axon was calculated, with ROIs weighted according to their size (that is, pixel number); 3) detection of significant Ca²⁺ transients (that is, transients larger than three standard deviations of the noise (that is, baseline $F$ values)). We then produced Ca²⁺ activity maps across all spatial locations and laps for each CA1 pyramidal cell and EC3 axon, using only those recording epochs, during which the animal was running (velocity >2 cm s⁻¹). These activity maps were generated by first dividing the length of the belt (that is, lap of 180 cm) into 50 spatial bins (3.6 cm each). For each spatial bin, the mean $\Delta f/f$ was calculated. All Ca²⁺ activity maps were then smoothed using a three-point boxcar, and for display purposes, aligned such that the opening of the valve (that is, reward delivery site) was located in either spatial bin 26 (Figs. 1–4 and Extended Data Fig. 2, data recorded in environment A) or spatial bin 24 (Figs. 5 and 6 and Extended Data Figs. 7, 9 and 11, data recorded in environment B, or when environments A and B are compared). Visual stimulation and reward locations are marked by arrows or dashed lines in all figures. All recorded laps were included, except for the data presented in Figs. 4a,b and 5d,e and Extended Data Figs. 7b–f and 9c–e (analysis of stochastic firing properties of EC3 axons), for which only laps 1–50 were used.

**CA1 place cell identification.** Many CA1 neurons were initially silent and acquired a place field suddenly during the recording sessions on day 0 or 1. Therefore, we first identified for each CA1 neuron a potential place field onset lap ('induction lap'). A place field onset was defined as a lap with a spatial bin with significant Ca²⁺ activity (greater than three standard deviations of the noise) in the neuron's eventual place field (defined as locations with contiguous activity >20% of peak mean $\Delta f/f$) in lap X, and the presence of spatial bins with significant Ca²⁺ activity in the neuron's eventual place field in two out of the five following laps (lap X + 1 to lap X + 6). If more than one lap per neuron fitted these criteria, we selected the first one, unless the field that was generated was weak and disappeared for more than 20 laps at some point during the recording. Only laps following the induction lap (that is, lap X) were used to determine whether a neuron was considered a place cell. Whether a CA1 neuron exhibited a spatially modulated field was defined by the amount of spatial information its activity provided about the linear track position (>95% confidence interval of the shuffled spatial information values) and by its reliability (significant activity in more than 30% of the laps following the induction lap). For each neuron, the spatial information, SI, was computed as described previously[55]:

$$\mathrm{SI} = \sum_i \lambda_i \log_2 \frac{\lambda_i}{\lambda} P_i$$

in which $P_i$ is the probability of occupancy in spatial bin $i$, $\lambda_i$ is the smoothed mean activity level ($\Delta f/f$) while occupying bin $i$, and $\lambda$ is the overall mean activity level ($\Delta f/f$). This value was compared to 100 shuffles of the activity (each shuffle was generated by circularly shifting the fluorescence trace by at least 500 frames, then dividing the fluorescence trace into six chunks and permuting their order). If the observed information value exceeded the 95% confidence interval of the shuffled information values, its field was considered spatially modulated. Neurons with no significant activity in any of the laps ('silent neurons') were not included in this analysis. The place field width was quantified as the number of consecutive spatial bins of 3.6 cm, for which the mean $\Delta f/f$ exceeded 20% of the peak $\Delta f/f$ value. Only one place field per neuron was included in our analyses.

**Behavioural data quantification.** As in the $Ca^{2+}$ imaging data analysis, the running and licking behavioural maps were generated by first dividing the length of the belt (that is, lap of 180 cm) into 50 spatial bins (3.6 cm each). Then, the lick rate (licks per second, Hz) and the mean velocity (cm s$^{-1}$) were calculated for each spatial bin.

**Velocity correlation of EC3 axons.** To categorize axons as significantly positively or negatively correlated with speed, Pearson's correlation coefficient (MATLAB function corr) was calculated between the mean $\Delta f/f$ $Ca^{2+}$ activity and the mouse's velocity (maps of 50 spatial bins per lap were used).

### Computational model
All computational modelling was performed in IGOR 8.04. A total of 2,000 two-state Markov chains, 510 s in duration, were generated using the transition probabilities shown in the matrix in Extended Data Fig. 7g to simulate 50 laps each 10 s in duration with a time step of 0.1 s (5,100 time steps total). Each of these chains simulated the persistent firing activity of a single EC3 neuron, for which state 0 was inactive and state 1 was active. The Markov chains were produced by randomly sampling numbers from a uniform distribution between 1 and 1,000 on each time step. At each time step, a chain transitioned from inactive to active if the sampled number was less than or equal to ($P_{01} \cdot 100$). Likewise, a chain transitioned from active to inactive if a randomly sampled number was less than or equal to ($P_{10} \cdot 100$). This produced exponentially distributed active and inactive times with means ($\tau_{on}$ and $\tau_{off}$), as expected from the transition probabilities (Extended Data Fig. 7j). For example, the probability of a chain transitioning from inactive to active during one time step ($\Delta t$) is $P = P_{01} \cdot \Delta t$, giving a mean inactive time of $\tau_{inactive} = \Delta t/P$ or $1/P_{01}$ (ref. [56]). Each of the fifty 100-time-step sections of the chains were averaged and smoothed with a three-point boxcar. The initial 100 points of each chain were not used, to allow proper initialization. For the activity versus position plots, the average activity in each spatial bin was calculated using the actual mean running speed of the animals.

In addition to this set of chains using static or homogeneous transition probabilities, we also used two other conditions. In these conditions, we were attempting to simulate the presence of two populations of chains, one that was purely homogeneous without any changes in their transition probabilities and a second population that had transition probabilities that were sensitive to the environment. We used two pieces of the EC3 data to direct our manipulations. The first was the median cell–cell correlations, and the second was the spatial distribution of peak activity (Extended Data Figs. 7–9). Thus, for the uniform track used in environment A, we altered the transition probabilities in the same proportion of chains uniformly across the lap (Extended Data Fig. 8a). For these conditions, at each of the 100 time steps, $P_{01}$ was step increased from 0.04 to 0.20 for 1 s (10 time steps) in 14 chains for a total of 1,400 chains in which the activation transition probability was adjusted. In the remaining 600 chains, $P_{01}$ was not changed (homogeneous condition). To simulate the nonuniform environment, we chose to manipulate the number of chains with increased $P_{01}$ around 40 cm such that the final fraction of chains

with peak activity around the light position was increased approximately threefold (Extended Data Fig. 9g). To do so, we used a similar procedure as above, except the number of chains with increased $P_{01}$ around 40 cm was elevated according to the density plot in Extended Data Fig. 9. The additional chains around the light stimulus were taken from the unmodulated pool, which was reduced to a total of 150 chains. Although this alone increased the median cell–cell correlations, it was also necessary to slightly increase the activation transition probability in all chains ($P_{01}$ stepped from 0.04 to 0.28 for 1 s) to approach the elevated median correlations observed in the data. To compare the correlation between odd and even laps correlations, a population of only 1,000 chains was used (but with all the same proportions) to simulate the experimental conditions more accurately. To produce the nonuniform distributions of spatial selectivity and odd–even correlations, we increased $P_{01}$ about threefold for about 100 chains around the appropriate position (see Extended Data Figs. 8 and 9).

These chains were used to predict the spatial plateau probability profile in a population of postsynaptic CA1 neurons (Extended Data Fig. 8i–k). To do this, we randomly selected 100 of the 2,000 chains and summed them (this represents 5% of the total 'input' population). We chose 2,000 because this is approximately the number of stratum lacunosum-moleculare synapses on CA1 pyramidal neurons, and 5% seemed a reasonable fraction of active inputs. We have altered this number between 2.5 and 10% and found the results to be consistent between 5 and 10%. Next, feedforward inhibition was simulated simply as the sum of all 2,000 chains scaled by the appropriate fraction (that is, multiplied by 0.05), and this waveform was subtracted from the sum of the 100 'excitatory EC3 inputs'. This procedure was repeated 10,000 separate times to mimic postsynaptic integration in a large population of CA1 neurons. Finally, a threshold was set on the basis of the observed fraction of the total CA1 population to generate new place fields during a session (20–25%). The fraction of 'CA1 neurons' that crossed the threshold (our proxy for plateau initiation probability) was calculated as the total number of threshold crossings in 30 spatial bins divided by the total number of 'neurons' (10,000) using the actual running speed profile of the animals or a constant speed profile of 18 cm s$^{-1}$ to determine the dwell time in each bin.

### Statistical methods
The exact sample size ($n$) for each experimental group is indicated in the figure legend or in the main text. No statistical methods were used to predetermine sample sizes, but our sample sizes are similar to those reported in previous publications[5,6,28,57] using a similar behavioural task and are guided by the expected number of active neurons or axons that can be imaged with the two-photon microscope in awake, behaving mice. In some cases, when data distribution was assumed, but not formally tested, to be normal, data were analysed using two-tailed paired or unpaired $t$-tests, as stated in the text or figure legends. Where indicated, Pearson's correlation coefficients were computed using the corr function in MATLAB. The corr function computed the $P$ values for Pearson's correlation using a Student's $t$ distribution for a transformation of the correlation. Experiments were randomized by randomly assigning littermate mice to the experimental groups. Data analyses were not performed blind to the experimental conditions but were analysed automatically, without consideration of trial conditions or experimental groups. Unless indicated otherwise in the figure caption, data are shown as mean ± s.e.m.

### Reporting summary
Further information on research design is available in the Nature Research Reporting Summary linked to this article.

## Data availability
The data supporting this study's findings are available from the corresponding author upon request. Source data are provided with this paper.

## Code availability

The code that supports the results of this study is available through a GitHub repository (https://doi.org/10.5281/zenodo.6998795).

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

**Acknowledgements** We thank R. Chitwood for technical assistance. We thank R. Chitwood, S. Romani and N. Spruston for useful discussions. This work was supported by the Howard Hughes Medical Institute and the Cullen Foundation.

**Author contributions** C.G. and J.C.M. designed the research. C.G. performed in vivo recordings. C.G. and J.C.M. analysed the experimental data. J.C.M. designed and implemented the computational model. C.G. and J.C.M. wrote the manuscript.

**Competing interests** The authors declare no competing interests.

**Additional information**
**Correspondence and requests for materials** should be addressed to Jeffrey C. Magee.

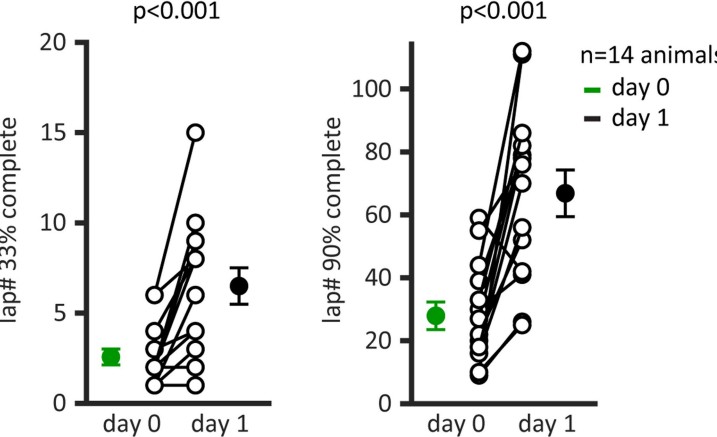

**Extended Data Fig. 1 | Time course of the CA1 place cell representation development.** Shown are the numbers of laps required to build 33% (left) and 90% (right) of the CA1 place cell representation on day 0 and day 1 (33%: n = 14; two-tailed paired *t*-test, day 0 vs. day 1, p = 3.34E-4; 90%: n = 14; two-tailed paired *t*-test, day 0 vs. day 1, p = 2.61E-4. The open circles show individual animals, the filled circles the mean. Data are shown as mean +/− SEM.

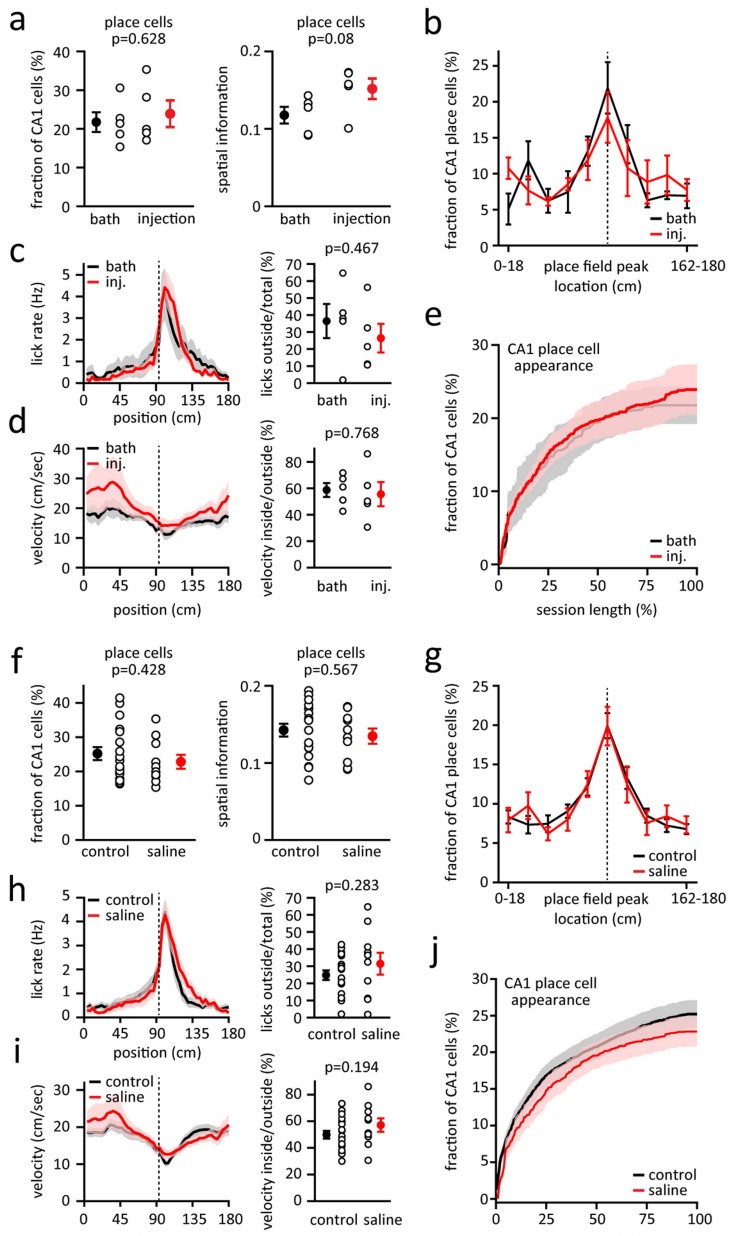

**Extended Data Fig. 2 | Bath application and injection of sterile saline 0.9% do not impact hippocampal activity and behavior. a–e**, Comparison of bath application (black, n = 5 animals) and injection (red, n = 5 animals) of sterile saline 0.9% in environment A. **a**, Left: Fraction of CA1 cells that are spatially modulated (two-tailed unpaired *t*-test, bath vs. injection, p = 0.628). Right: Mean place cell spatial information content per animal (two-tailed unpaired *t*-test, bath vs. injection, p = 0.08). **b**, Fraction of CA1 place cells as a function of place field peak location. The track is divided into ten spatial bins of 18 cm. **c**, Left. Mean lick rates for bath application and injection of sterile saline 0.9%. Right: Mean number of licks outside the reward zone (from 14 cm before to 36 cm after the reward) divided by the total number of licks (two-tailed

unpaired *t*-test, bath vs. injection, p = 0.467). **d**, Left: Mean running profile for bath application and injection of sterile saline 0.9%. Right: Minimum velocity inside the reward zone divided by mean velocity outside the reward zone (two-tailed unpaired *t*-test, bath vs. injection, p = 0.768). **e**, Time course of CA1 place cell appearance for bath application and injection of sterile saline 0.9%. In panels a, c, d the open circles show individual animals, the filled circles the mean. **f–j**, Same as **a–e** for comparing control condition (black, no saline, n = 18 animals) and sterile saline 0.9% application via bath application and injection (red, n = 10 animals) in environment A. Black dashed lines depict the reward location. Data are shown as mean +/− SEM.

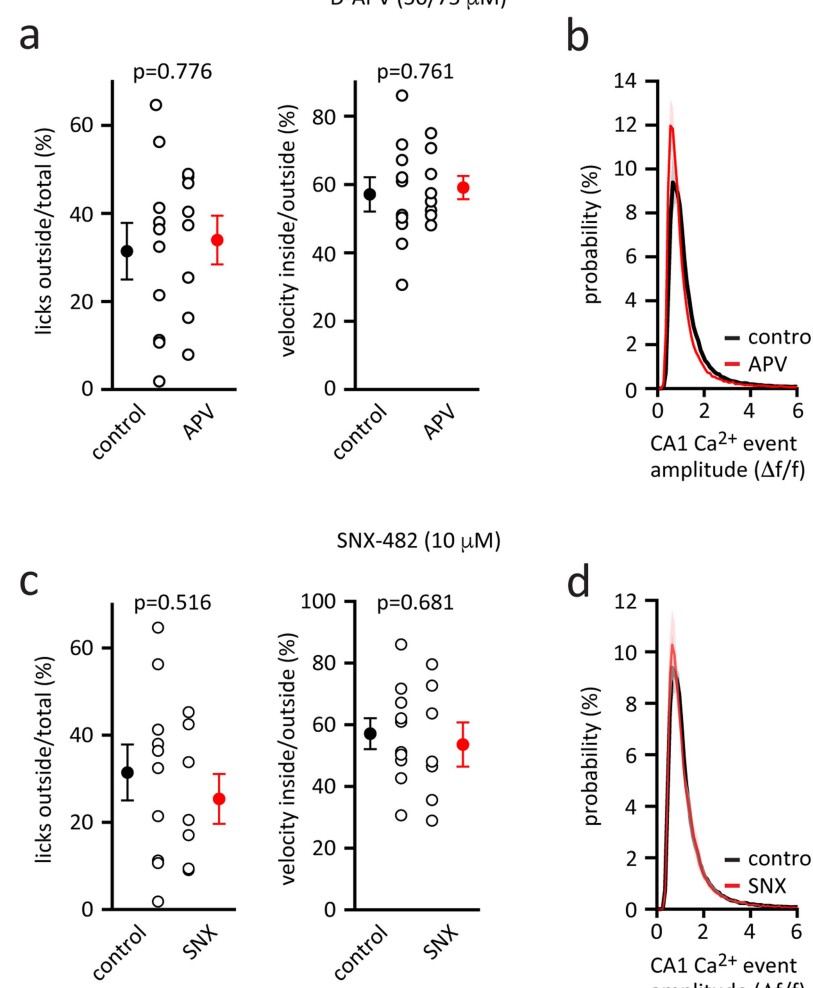

**Extended Data Fig. 3 | Effect of locally applied BTSP blockers on behavior and CA1 Ca²⁺ event amplitude. a-b**, Effect of NMDA receptor antagonist, D-AP5 (50 or 75 μM). Black: Control (n = 10 animals). Red: D-AP5 (n = 8 animals). **a**, Left. Mean number of licks outside the reward zone (from 14 cm before to 36 cm after the reward) divided by the total lick number. Right: Mean velocity inside the reward zone divided by mean velocity outside the reward zone. The open circles show individual animals, the filled circles the mean. **b**, Distribution of CA1 Ca²⁺ event amplitudes recorded. **c-d**, Effect of Ca²⁺ channel antagonist, SNX-482 (10 μM). Black: Control (n = 10 animals). Red: SNX-482 (n = 7 animals). Panels same as **a-b**. Two-tailed unpaired *t*-test were used, and data are shown as mean +/– SEM.

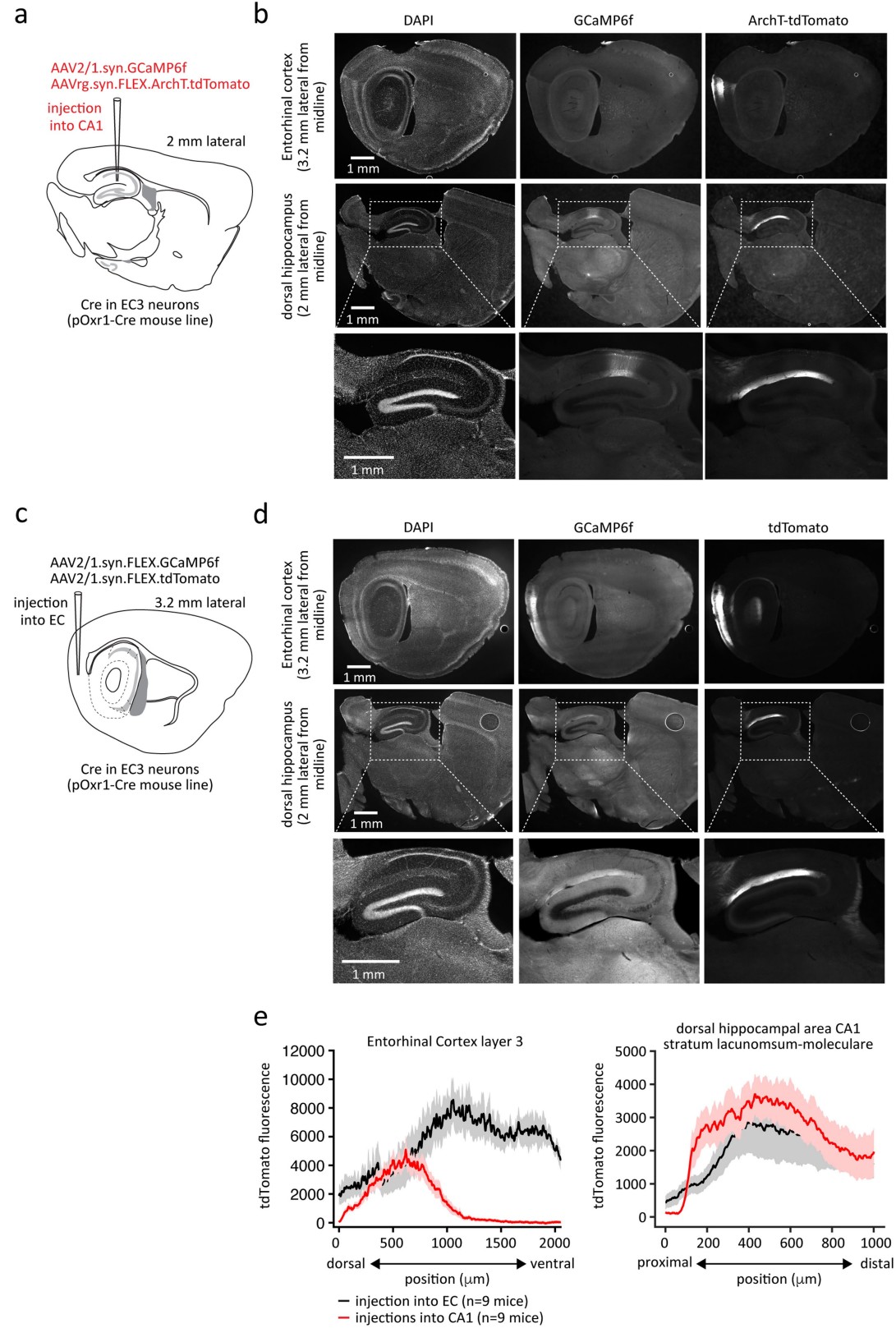

**Extended Data Fig. 4** | See next page for caption.

**Extended Data Fig. 4 | Histological analyses of virally injected pOxr1-Cre mice. a**, First experimental strategy of co-injecting AAVrg (retrograde).syn. FLEX.ArchT.tdTomato and AAV2/1.syn.GCaMP6f injections into dorsal CA1 (dCA1). **b**, Representative images of sagittal slices (50 μm thickness) from one animal. Shown are the entorhinal cortex (top row), the dCA1 (middle row), and a magnified view of the dCA1 (lower row). Left column: blue channel/DAPI. Middle column: green channel/GCaMP6f. Right column: red channel/ tdTomato. **c-d**, same as **a-b** for the second experimental strategy of co-injecting AAV2/1.syn.FLEX.GCaMP6f and AAV2/1.syn.FLEX.tdTomato into the entorhinal cortex. The mouse brain sections in panels a and c have been reproduced with permission from ref. [52]. **e**, Raw tdTomato fluorescence values in layer 3 entorhinal cortex (from dorsal to ventral, 3.2 mm lateral from the midline) and in stratum lacunosum-moleculare of dorsal CA1 (from proximal to distal, 2 mm lateral from the midline), measured in 9 pOxr1-Cre mice each. Black: AAV2/1.syn.FLEX.GCaMP6f and AAV2/1.syn.FLEX.tdTomato into the entorhinal cortex. Red: AAVrg (retrograde).syn.FLEX.ArchT.tdTomato and AAV2/1.syn.GCaMP6f into dCA1. Shown are mean +/− SEM, calculated from values obtained using the line profile function in ImageJ (Version 2.0.0).

## a Entorhinal cortex LFP without and with optogenetic perturbation

No light

+594 nm, 40 Hz stimulation, 5-10 mW at the fiber tip (in air)

**Extended Data Fig. 5 | Optogenetic perturbation of entorhinal cortex layer 3 neurons via Archaerhodopsin T (ArchT). a**, Entorhinal cortex LFP without (black) or with (red) optogenetic perturbation via ArchT expression in layer 3 (n = 6 sessions from 4 animals). Top. Raw LFP signal recorded in the same animal. Bottom: Power spectral density analysis (thin lines: individual sessions; thick lines: mean) and theta-to-gamma-ratio (two-tailed $t$-test, p = 0.0139). **b–e**, Basic CA1 place cell features and the behavior are indistinguishable in tdTomato control (black, n = 6) and ArchT (red, n = 8) animals. **b**, Left: Fraction of CA1 cells that are spatially modulated (two-tailed unpaired $t$-test, p = 0.886).

Right: Mean place cell spatial information content per animal (two-tailed unpaired $t$-test, p = 0.627). **c**, Distribution of CA1 $Ca^{2+}$ event amplitudes recorded. **d**, Time course of CA1 place cell appearance for tdTomato control and ArchT animals. **e**, Left: Mean number of licks outside the reward zone (from 14 cm before to 36 cm after the reward) divided by the total number of licks (two-tailed unpaired $t$-test, p = 0.603). Right: Mean velocity inside the reward zone divided by mean velocity outside the reward zone (two-tailed unpaired $t$-test, p = 0.697). In panels a, b, and e, the open circles show individual animals, the filled circles the mean. Data are shown as mean +/− SEM.

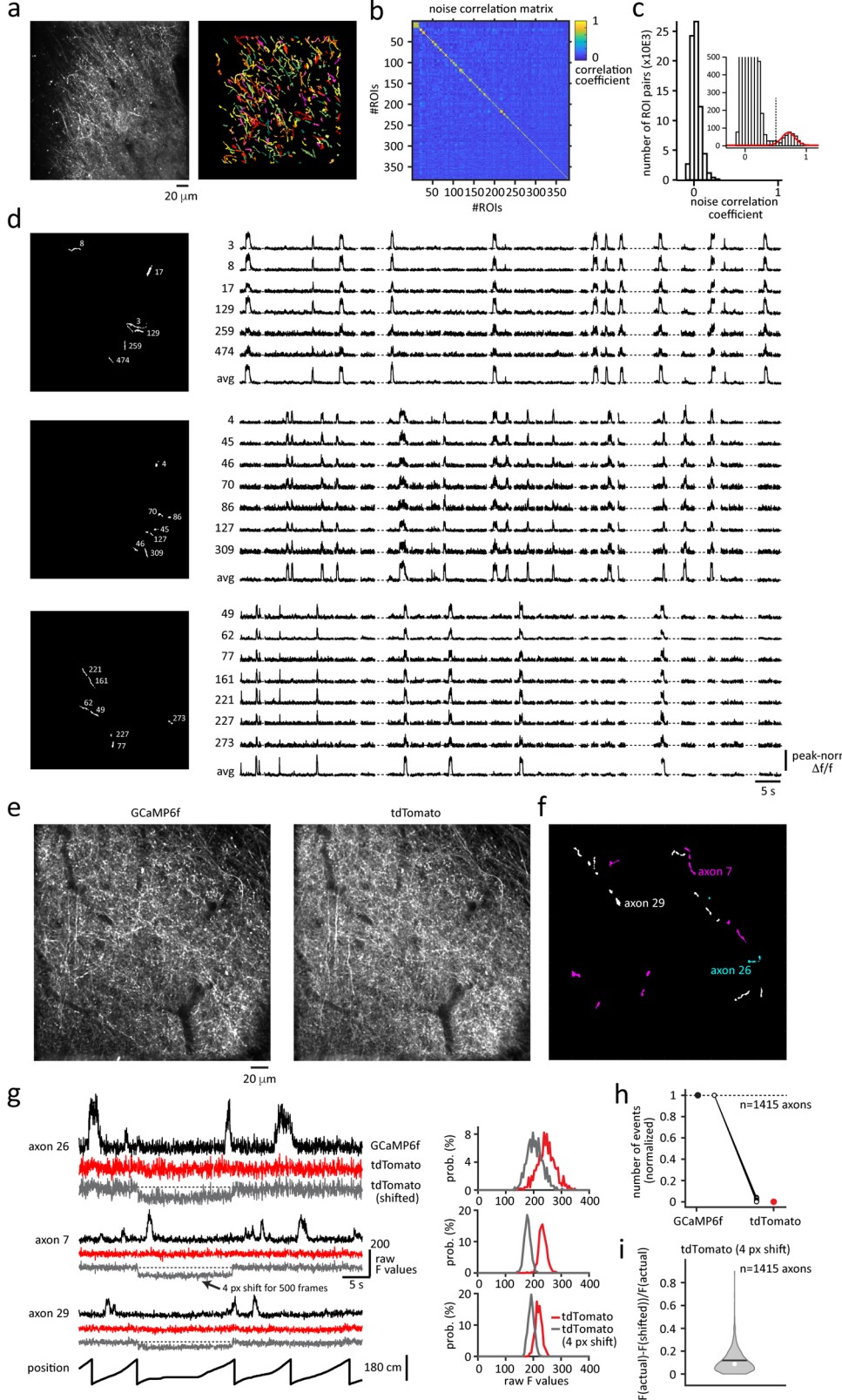

**Extended Data Fig. 6** | See next page for caption.

**Extended Data Fig. 6 | Entorhinal cortex layer 3 axon imaging. a–c**, Analysis pipeline to identify ROIs that belong to the same axon. Left. Representative single-plane, two-photon, time-averaged image showing expression of GCaMP6f in EC3 axons in a single animal. Right. All axonal regions of interest (ROIs), as identified by Suite2p in the image from **a**. Colors are assigned randomly. **b**, Noise correlation matrix for all axonal ROIs identified in this animal (n = 369). ROIs are sorted so that highly correlated ROIs are clustered. **c**, Histogram showing the noise correlation coefficient distribution for all ROI pairs from this animal. Inset shows magnified view of second, distinct, histogram peak (red line depicts gaussian fit of this second peak). All ROIs with a noise correlation coefficient value > 0.5 were assumed to belong to the same axon and subsequently combined into a single compound ROI. **d**, 3 example axons. Left: white areas depicting individual ROIs, which are assumed to belong to a single axon. Right. Normalized Δf/f traces for these ROIs (numbers correspond to the masks on the left) and for the resulting single axon (avg). The gaps represent epochs during which Ca²⁺ signal was not recorded (e.g., because the animal was stationary). **e–i**, Simultaneous imaging of GCaMP6f and tdTomato in EC3 axons as control for z-motion. **e**, Representative single-plane, two-photon, time-averaged image showing expression of the GCaMP6f (green channel) and tdTomato (red channel) in EC3 axons in a single animal. **f**, Colored areas depict three individual axons in the image shown in **e**, as identified by the noise correlation analysis. **g**, Left. Raw fluorescence traces (black: GCaMP6f, red: tdTomato) for the three axons. The grey traces are obtained by shifting all ROIs belonging to the axon for 500 frames by 4 pixels (px) in the x-dimension. The position signal at the bottom indicates epochs of varying running speeds. Regardless of the animal's velocity, the tdTomato signal remains stable. Right: Raw F value histograms of the 500-frame period where the axonal ROIs are shifted. **h**, Number of events (normalized) per axon (n = 1415 axons from 14 animals). The open circles show individual animals, the filled circles the mean (paired two-tailed *t*-test, p = 0). **i**, Difference between recorded raw tdTomato fluorescence values and the fluorescence values when ROIs are artificially shifted by 4 pixels in the x dimension. The difference is shown as a fraction of the actual value. The white dot marks the median, the black line the mean of the distribution. A 4-pixel shift (-2 μm) would have caused a detectable (-10%) change in the tdTomato fluorescence. If not otherwise indicated, data are shown as mean +/− SEM.

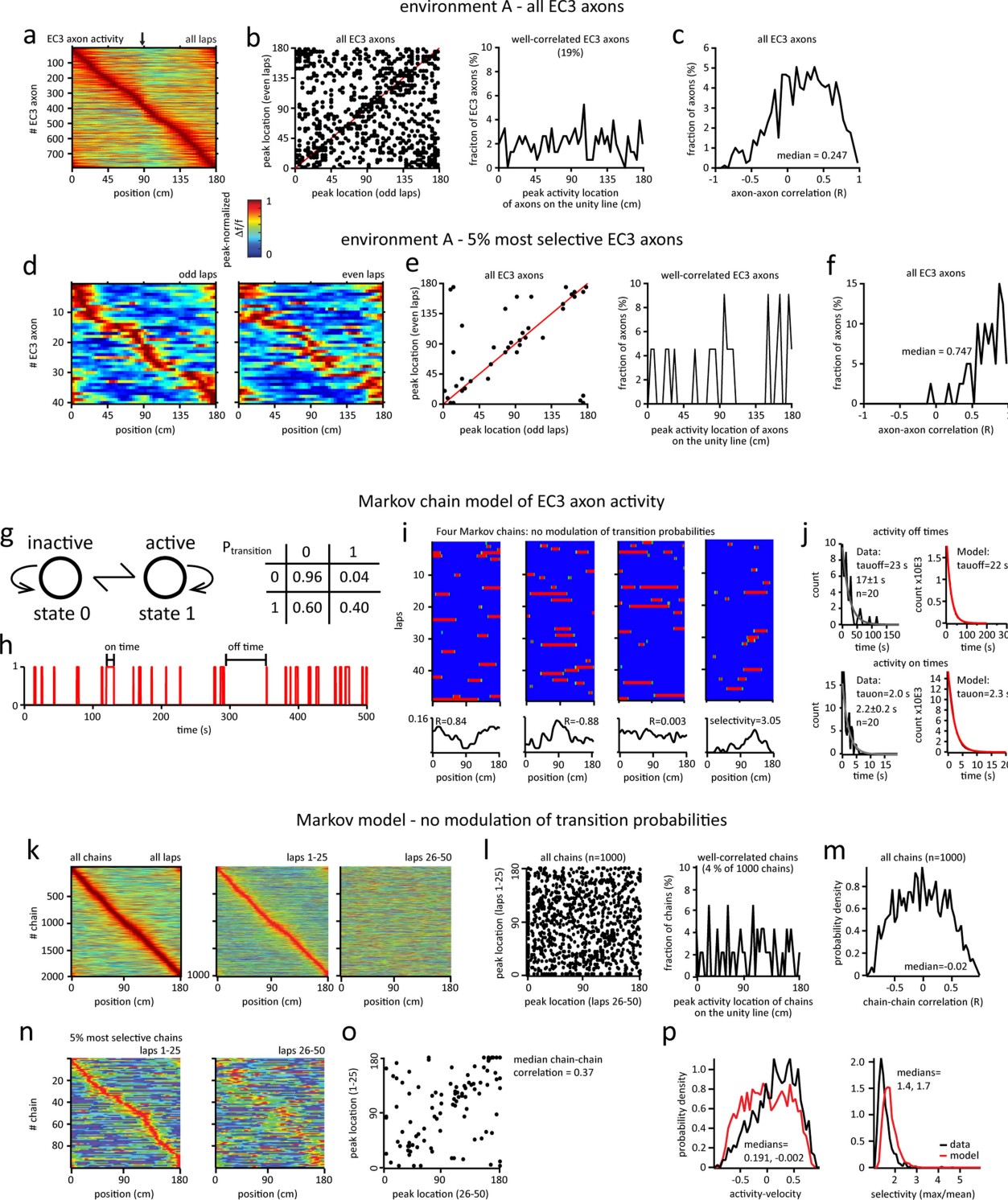

**Extended Data Fig. 7** | See next page for caption.

**Extended Data Fig. 7 | Modeling of EC3 axon activity. a**–**f**, Characterization of EC3 axonal activity, recorded in environment A. **a**, Peak-normalized mean Δf/f across space for all entorhinal cortex layer 3 (EC3) axons recorded in the environment (env) A (n = 792 axons in 7 animals). EC3 axons are ordered according to their peak activity location. **b**, Left: Scatter plot showing axonal activity peak locations for averages made from odd and even laps. The unity line is depicted in red. Right: Histogram of peak activity locations of well-correlated axons located on the unity line. **c**, Distribution of Pearson's correlation coefficients for axon-axon comparisons from data in **b**. **d**, Peak-normalized mean Δf/f across space for the 5% most selective axons recorded in env A. Colormaps for odd and even laps are shown separately. EC3 axons are ordered according to their peak activity location in the odd laps. **e-f**, Same as panels **b**–**c**, but only the 5% most selective axons are included. **g**–**o**, Markov chain model of EC3 axon activity. **g**, Individual EC3 axon activity was modeled as a two-state Markov chain with base transition probabilities as shown in the matrix. Each chain, 2000 in total, was generated as transitions from 0 (inactive) to 1 (active) for 50 laps with a single lap duration of 10 s. **h**, Representative chain showing the transitions used to calculate active (on) times and inactive (off) times. **i**, Activity heat maps for 50 laps (10 s each) for four different chains. From left to right, chain showing high positive velocity correlation, chain showing high negative velocity correlation, chain showing no correlation and chain showing strong selectivity (maximum amplitude/mean amplitude of average trace. Average traces shown below. **j**, Activity times are exponentially distributed for real EC3 single axons (left plots, black, 20 axons with highest number of events) and the population of model chains (right plots, red, all chains). **k**, Chains with static (unmodulated) transition probabilities. From left to right: heat maps of peak-scaled mean chain activity for 50 laps from population of 2000 chains plotted in space. Average activity for the first 25 laps for subset of 1000 chains. Average activity for the last 25 laps for same chains. **l**, Left: plot of peak locations for average activity from first 25 laps versus last 25 laps. Right: Density of well-correlated chains (chains whose peak locations were within 10 cm in laps 1–25 and 26–50). **m**, Distribution of Pearson's correlation coefficients for chain-chain comparisons from data in **l**. **n**–**o**, Same as panels **k**–**l**, but only the 5% most selective chains are included. **p**, Distributions of activity-velocity correlation coefficients (left) and selectivity indices (right) for population of EC3 axons (black) and unmodulated model data (red). Medians are listed.

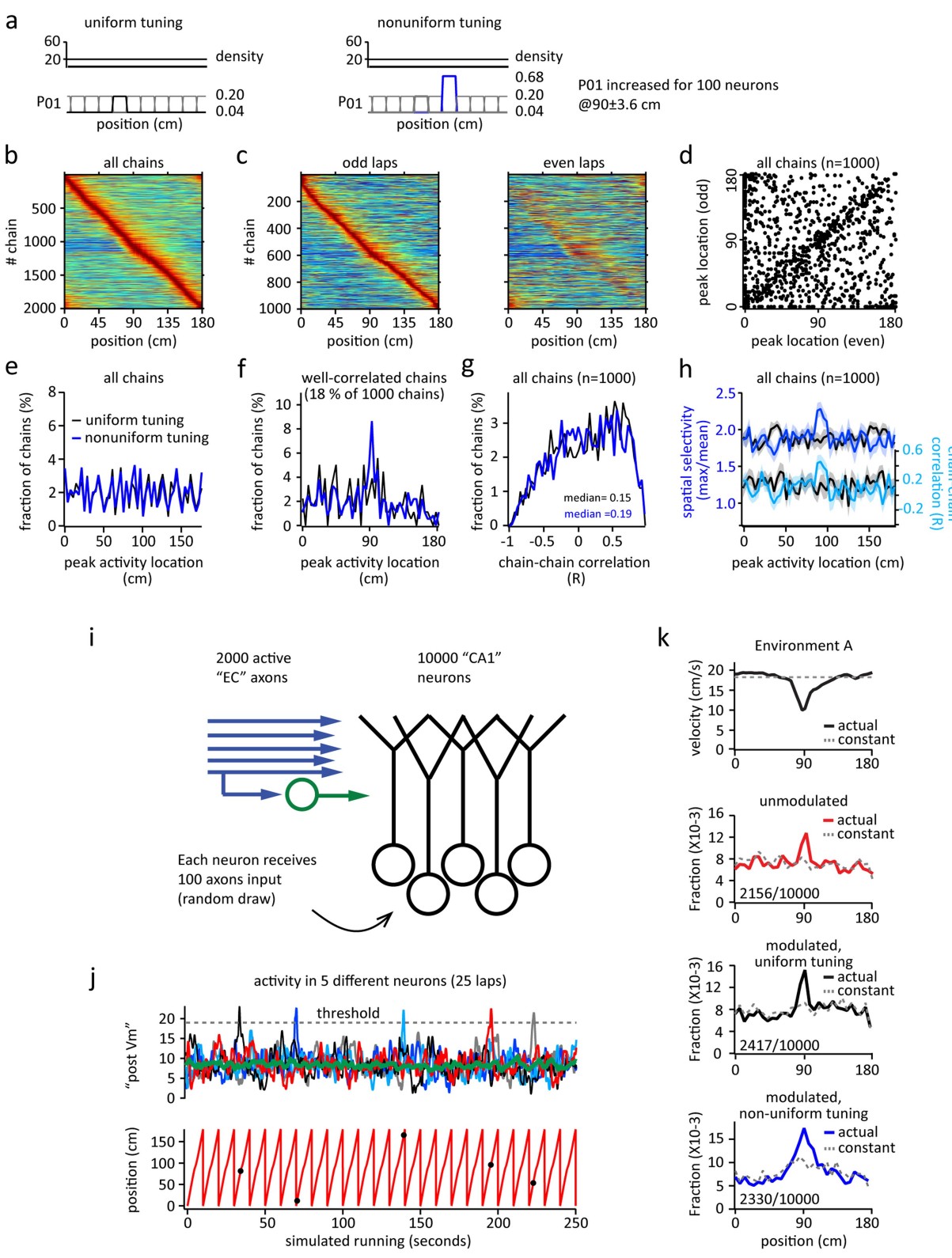

# Markov model-- Environment A: (70% modulated; 30% unmodulated)

**Extended Data Fig. 8** | See next page for caption.

**Extended Data Fig. 8 | Analysis of the Markov chain model of EC3 axon activity and predictions for environment A. a**, Transition probabilities and chain density, where $P_{0,1}$ was modulated in 1400 chains by a 1 s step that moved iteratively across the lap. Left. Modeling without tuning added. Right. Modeling with tuning added. **b**, Peak-normalized mean chain activity for 50 laps from population of 2000 chains plotted in space. **c**, Peak-normalized activity across space for 1000 modeled chains. Color plots for odd and even laps are shown separately. Chains are ordered according to their peak location during the odd laps. **d**, Scatter plot showing chain activity peak locations for averages made from odd and even laps. **e**, Histogram of peak activity for all chains. **f**, Histogram of peak activity locations of well-correlated chains. Peak locations on even trials were within 10 cm of odd trials. **g**, Distribution of Pearson's correlation coefficients for chain-chain comparisons. **h**, Spatial selectivity (max/mean) and chain-chain correlation (odd vs. even laps) as a function of peak activity location. **i**, Model of impact on postsynaptic neurons. One hundred (5%) 50-lap chains are randomly sampled from the total of two thousand and summed 10,000 separate times to simulate postsynaptic summation in a population of 10,000 CA1 dendrites. In addition, an average activity from all 2000 (scaled appropriately: 0.05x) was subtracted from each summed chain to mimic feedforward inhibition. **j**, The activity in five representative postsynaptic neurons (100 summed chains) is shown for 25 consecutive laps. Inhibitory trace (green). Threshold, set so that 20–25% of postsynaptic neurons cross, is demarcated by dashed line. Below is the position of the mouse from actual data (black circles indicate location in space of threshold crossing). **k**, Plots simulating environment A showing from top to bottom: actual running speed profile in space (black) and a trace simulating a constant running speed (gray dashed). The fraction of postsynaptic neurons with threshold crossings for the unmodulated condition, the modulated condition without tuning added, and the modulated condition with tuning added vs. position on the track. If not indicated otherwise, data are shown as mean +/− SEM.

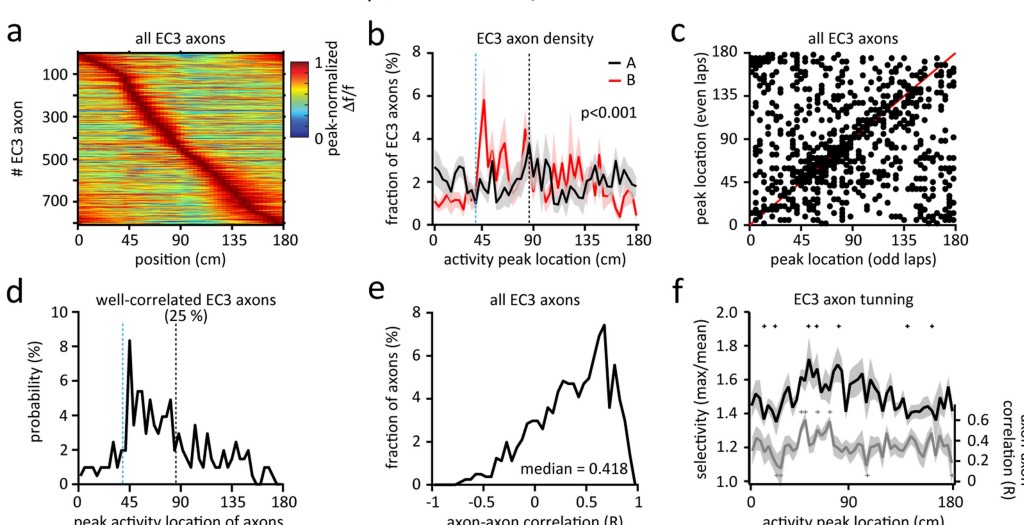

Experimental data / Environment B

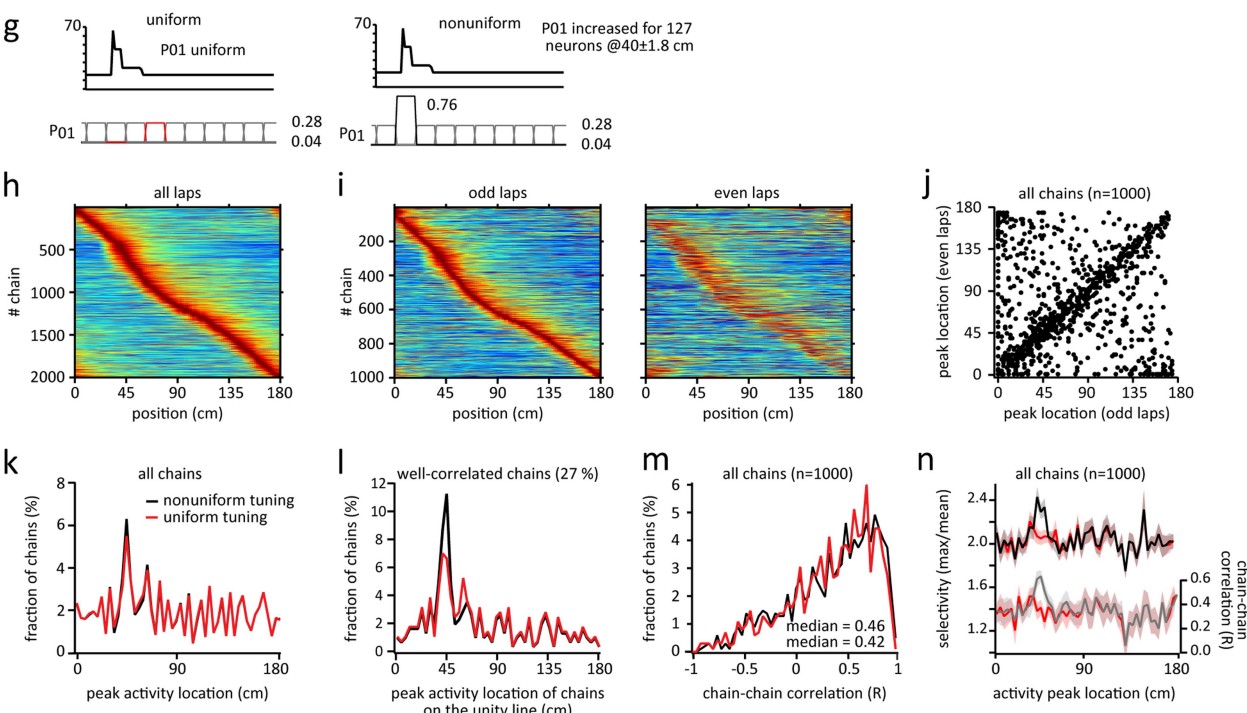

Markov model / Environment B: (92.5% modulated; 7.5% unmodulated)

**Extended Data Fig. 9 | Analysis of EC3 axon activity in environment B and the predictions of the Markov chain model. a-f** Characterization of EC3 axonal firing in environment (env) B. **a**, Normalized mean Δf/f across space for EC3 axons (n = 808, n = 8 animals) in env B. EC3 axons are ordered according to their peak activity location. **b**, Fraction of EC3 axons as a function of their peak activity location (env A: n = 7, black; env B: n = 8, red; chi-square test, df = 49, p = 3.79E-36). The track is divided into 50 spatial bins of 3.6 cm each. **c**, Plot of peak locations for average activity from even laps versus odd laps in env B. **d**, Histogram of peak activity locations of well-correlated axons, located on the unity line. **e**, Distribution of Pearson's correlation coefficients for axon-axon comparisons from data in **c**. **f**, Spatial selectivity (max/mean) and axon-axon correlation (odd vs. even laps) as a function of peak activity location. The plus symbols indicate values that lie outside 95% CI generated from 1000 shuffles of n = 808 data points. **g-n**, Modeling of EC3 axon activity in environment B.

**g**, Transition probabilities and chain density, where $P_{0,1}$ was modulated in 1400 chains by a 1 s step that moved iteratively across the lap. Left. Modeling without tuning added. Right. Modeling with tuning added. **h**, Heat maps of normalized mean chain activity for 50 laps from population of 2000 chains plotted in space. **i**, Peak-normalized activity across space for 1000 modeled chains. Color plots for odd and even laps are shown separately. Chains are ordered according to their peak location during the odd laps. **j**, Scatter plot showing chain activity peak locations for averages made from odd and even laps. **k**, Histogram of peak activity for all chains. **l**, Histogram of peak activity locations of well-correlated chains. Peak locations on even trials were within 10 cm of odd trials. **m**, Distribution of Pearson's correlation coefficients for chain-chain comparisons from data in **j**. **n**, Spatial selectivity (max/mean) and chain-chain correlation (odd vs. even laps) as a function of peak activity location. If not indicated otherwise, data are shown as mean +/- SEM.

a

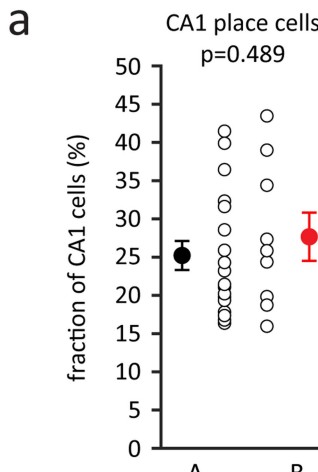

b

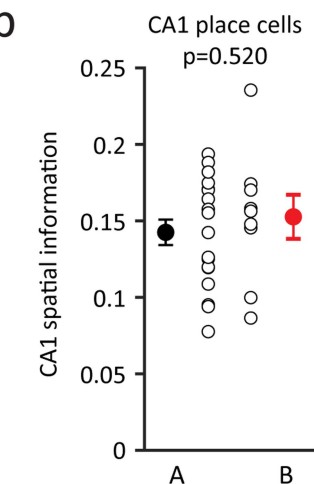

**Extended Data Fig. 10 | Basic CA1 place cell features are similar in environments A (somatosensory cues on the belt, n = 18 animals) and B (no cues on the belt, visual stimulus only, n = 9 animals). a**, Fraction of CA1 cells that are spatially modulated (n = 18 and 9 mice, respectively, two-tailed *t*-test, p = 0.489). **b**, Mean place cell spatial information content per animal (n = 18 and 9 mice, respectively, two-tailed *t*-test, p = 0.520). The open circles show individual animals, the filled circles the mean. Data are shown as mean +/− SEM.

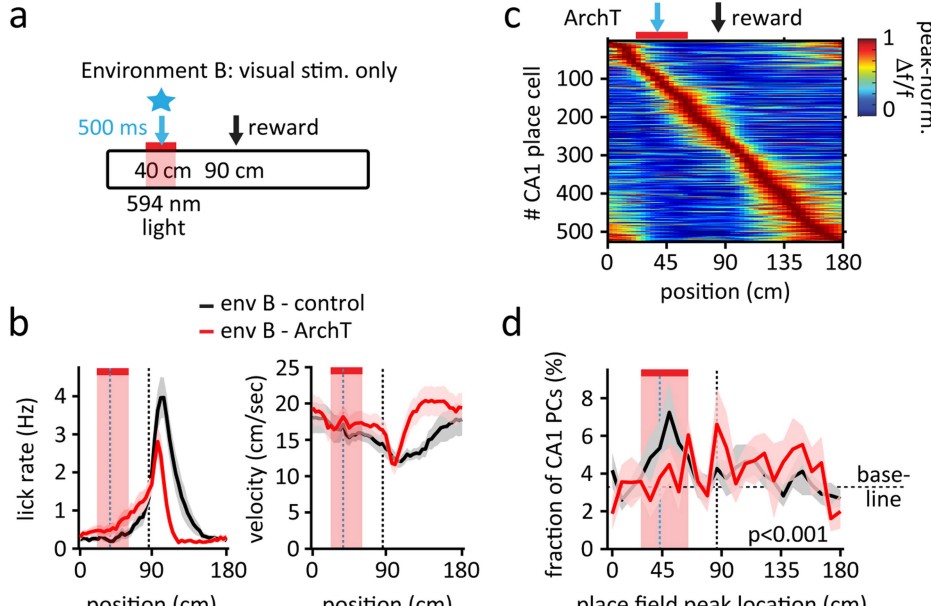

**Extended Data Fig. 11 | Ipsilateral optogenetic perturbation of entorhinal cortex layer 3 (EC3) neuronal activity prohibits the development of the visual stimulus overrepresentation in environment B.** Black: control animals without any fluorescent protein (FP) expression (n = 9 animals, data also shown in Fig. 5), Red: ArchT (n = 9 animals). **a**, The environment (env) B (red) involves a blank belt and a visual stimulus (blue LED flashes, 10 Hz,

500 ms) 50 cm before the single, fixed, reward (black arrow). Red bar marks light-on locations. **b**, Mean lick rate (left) and mean running profile (right) in env B. **c**, Peak-normalized mean Δf/f across space for all CA1 place cells (PCs), recorded in ArchT-expressing mice. **d**, Fraction of CA1 PCs as a function of place field peak location (chi-square test, df = 24, p = 1.92E-5). The track is divided into 25 spatial bins of 7.2 cm each. Data are shown as mean +/− SEM.

# Reporting Summary

## Statistics

For all statistical analyses, confirm that the following items are present in the figure legend, table legend, main text, or Methods section.

| n/a | Confirmed | |
|---|---|---|
| ☐ | ☒ | The exact sample size (*n*) for each experimental group/condition, given as a discrete number and unit of measurement |
| ☐ | ☒ | A statement on whether measurements were taken from distinct samples or whether the same sample was measured repeatedly |
| ☐ | ☒ | The statistical test(s) used AND whether they are one- or two-sided *Only common tests should be described solely by name; describe more complex techniques in the Methods section.* |
| ☐ | ☒ | A description of all covariates tested |
| ☐ | ☒ | A description of any assumptions or corrections, such as tests of normality and adjustment for multiple comparisons |
| ☐ | ☒ | A full description of the statistical parameters including central tendency (e.g. means) or other basic estimates (e.g. regression coefficient) AND variation (e.g. standard deviation) or associated estimates of uncertainty (e.g. confidence intervals) |
| ☐ | ☒ | For null hypothesis testing, the test statistic (e.g. *F*, *t*, *r*) with confidence intervals, effect sizes, degrees of freedom and *P* value noted *Give P values as exact values whenever suitable.* |
| ☒ | ☐ | For Bayesian analysis, information on the choice of priors and Markov chain Monte Carlo settings |
| ☒ | ☐ | For hierarchical and complex designs, identification of the appropriate level for tests and full reporting of outcomes |
| ☐ | ☒ | Estimates of effect sizes (e.g. Cohen's *d*, Pearson's *r*), indicating how they were calculated |

*Our web collection on statistics for biologists contains articles on many of the points above.*

## Software and code

Policy information about availability of computer code

| Data collection | The Ca2+ imaging data was recorded using a National Instruments PXI system, controlled by ScanImage (R2015 and R2018, Vidrio). The behavioral data was acquired using a NI PCIe-6343 card, connected to a BNC-2090A rack-mountable break-out box, controlled by Wavesurfer (Version 0.982, open-access software, wavesurfer.janelia.org). The behavioral system was controlled via an Arduino microprocessor and custom-written Arduino and Matlab code. All histological images were acquired on the ZEISS Axio Zoom.V16 microscope equipped with ZEN 3.1 software. |
|---|---|
| Data analysis | To extract somatic Ca2+ signals of CA1 pyramidal neurons, videos were motion-corrected using SIMA (Version 1.3.2), neurons were manually drawn (using Image J version 2.0.0), and calcium traces across time were extracted using SIMA. To extract axonal EC3 Ca2+ signals, the automatic motion correction and ROI detection algorithm of the Suite2P (Version 0.6.16) analysis package was used. Further analyses of CA1 and EC3 activity were performed using custom functions written in MATLAB (Version 2019a). The modeling was performed in IGOR 8.04. Histological sections were analyzed using Image J's line plot function (Image J version 2.0.0). The code used to analyze the experimental data and perform the modeling will be available via a GitHub repository. |

For manuscripts utilizing custom algorithms or software that are central to the research but not yet described in published literature, software must be made available to editors and reviewers. We strongly encourage code deposition in a community repository (e.g. GitHub). See the Nature Portfolio guidelines for submitting code & software for further information.

## Data

Policy information about availability of data

All manuscripts must include a data availability statement. This statement should provide the following information, where applicable:

- Accession codes, unique identifiers, or web links for publicly available datasets
- A description of any restrictions on data availability
- For clinical datasets or third party data, please ensure that the statement adheres to our policy

The data that support the findings of this study are available from the corresponding author upon request.

# Field-specific reporting

Please select the one below that is the best fit for your research. If you are not sure, read the appropriate sections before making your selection.

☒ Life sciences ☐ Behavioural & social sciences ☐ Ecological, evolutionary & environmental sciences

For a reference copy of the document with all sections, see nature.com/documents/nr-reporting-summary-flat.pdf

# Life sciences study design

All studies must disclose on these points even when the disclosure is negative.

| Sample size | No statistical methods were used to predetermine sample sizes. The number of mice per group was determined by previous publications on a similar behavioral task than used in our study (refs. 5, 6, 27, 54), by the expected number of active neurons or axons that can be imaged with the two-photon microscope in awake behaving mice (refs. 5, 6, 27, see also Danielson et al, Neuron, 2016). The main effects were significant with the number of mice, neurons or axons in each group, and the effects were consistent across individual mice and neurons within each group, as evident by the presentation of individual data throughout the paper. |
|---|---|
| Data exclusions | Animals were excluded from further analyses for two reasons: 1) extensive z-motion did preclude imaging of the same population of neurons throughout the recording sessions; 2) an animal did run less than 20 laps per recording session of 45-60 minutes. |
| Replication | We used appropriate sample sizes and indicate those throughout the manuscript. All experiments were performed independently. Our main findings are maintained across all animals, and individual data points representing individual animals are shown for most of our analyses. We do not show individual data points in those plots that contains many bins, thus making it difficult to see. All replications were successful. |
| Randomization | Littermate GP5.17 or littermate pOxr1- Cre mice were used and randomly assigned to each experiment. For all manipulation experiments, we compared experimental group to the appropriate control group. For the hippocampal pharmacology experiment, GP5.17 littermate mice were used and were randomly assigned to the two experimental groups (APV/SNX vs. vehicle application). For the optogenetic experiments, pOxr1-Cre litter mate mice were used and randomly assigned to the two experimental groups (viral expression of ArchT-tdtomato or tdtomato). |
| Blinding | Experiments and data analyses were not performed blind to the experimental conditions. This was due to the fact that the experimenter also applied the experimental drug during the recording or performed the surgeries and was thus not be able to be blinded to the experimental conditions. All analyses were performed using automatized data analyses procedures. |

# Reporting for specific materials, systems and methods

We require information from authors about some types of materials, experimental systems and methods used in many studies. Here, indicate whether each material, system or method listed is relevant to your study. If you are not sure if a list item applies to your research, read the appropriate section before selecting a response.

### Materials & experimental systems

| n/a | Involved in the study |
|---|---|
| ☒ | ☐ Antibodies |
| ☒ | ☐ Eukaryotic cell lines |
| ☒ | ☐ Palaeontology and archaeology |
| ☐ | ☒ Animals and other organisms |
| ☒ | ☐ Human research participants |
| ☒ | ☐ Clinical data |
| ☒ | ☐ Dual use research of concern |

### Methods

| n/a | Involved in the study |
|---|---|
| ☒ | ☐ ChIP-seq |
| ☒ | ☐ Flow cytometry |
| ☒ | ☐ MRI-based neuroimaging |

# Animals and other organisms

Policy information about studies involving animals; ARRIVE guidelines recommended for reporting animal research

Laboratory animals

All experiments were performed in adult (older than P66 at the time of surgery) GP5.17 (n=52 mice, Janelia and Jackson Laboratories) or pOxr1-Cre (n=44 mice, Jackson Laboratories) mice of either sex. Animals were housed under an inverse 12-hour dark/12-hour light cycle (lights off at 9 am) in the Magee lab satellite facility with temperature (~21 degree Celsius) and humidity (~30-60 %) controlled.

Wild animals

this study does not involve wild animals.

Field-collected samples

this study does not involve field-collected samples.

Ethics oversight

All experiments were performed according to methods approved by the Janelia (Protocol 12-84 & 15-126) and the Baylor College of Medicine's (Protocol AN-7734) IACUC committees.

Note that full information on the approval of the study protocol must also be provided in the manuscript.

