## [Peer Review File · Nature]

Manuscript Title: Entorhinal cortex directs learning-related changes in CA1 representations

Reviewer Comments & Author Rebuttals

Reviewer Reports on the Initial Version:

Referees' comments:

Referee #1 (Remarks to the Author):

This paper studies the mechanisms of CA1 place code reorganization during learning, in particular for the enrichment of place cells around a reward location. The main finding is that behavioral timescale synaptic plasticity contributes to this reorganization and that BTSP in CA1 originates from EC3 synaptic inputs. I found the paper to be exciting in linking BTSP to a well known reshaping of the place code in the hippocampus, thus extending the previous findings from this group on the characterization of BTSP. I also found the mechanistic investigation of EC3's input to be exciting and important. I appreciated the combination of measurements of neural activity, manipulations of specific circuit elements, and computational modeling. Overall, I found the study to be well done and thorough and the paper to be well written and clearly presented. I had one comment that lessened my enthusiasm slightly, regarding the link of BTSP to the behavior, and a few specific comments.

1. The paper is introduced by talking about how learning underlies adaptive behavior and how learning of a reward site requires an over-representation of place cells near the reward site (first two sentences of the abstract). In this paper, disrupting the EC3 inputs to CA1 disrupts the over-representation near the reward site (Figure 3b-c). However, the behavior is not disrupted and appears similar to when EC3 inputs are intact (ED Figure 4e). This implies that BTSP can be disrupted and that the place cell reorganization can be prevented without any effect on the behavior. Thus, while there appears to be a tight link between BTSP, EC3 inputs, and the place cell enrichment near the reward site, there does not seem to be a link between any of these and the behavior itself. Related to this point, in the logic and model of Figure 4, the key determinant of the EC3 effect on place cell enrichment near the reward site seems to be the longer dwell time of the mouse near the reward site. If so, then the increase in dwell time near the reward site, which is a behavioral indicator of learning, precedes the BTSP events. Thus, the BTSP events might be causal for the place cell enrichment, but these events and the place cell enrichment would not be causal for the behavioral learning as they follow in time behaviors that are indicative of learning.

It would be helpful for the authors to comment on these points in the paper and to hear if they agree with these conclusions or have evidence to speak against my interpretation. If the authors agree with my conclusions, then I would advise lessening the emphasis in the introduction on the relationship to behavior. Instead, the link between the plasticity and the place cell enrichment seems strong and should be emphasized more clearly as the focus, instead of understanding plasticity that underlies behavior. In this case, the plasticity might not underlie the behavior that is studied. Also, if the authors agree with my conclusions, it would be helpful to add a brief discussion

of these points at the end of the paper.

2. Figure 4 was presented in a way that indicates that the dwell time was critical and that one could predict the BTSP events based on the product of the dwell time and dF/F , as presented in Figure 4e. However, this does not seem to be true for the experiments in Figure 5. The EC3 axons have higher activity at the blue light time and longer dwell times near the reward. I therefore anticipate that the product of dF/F and dwell time would be similar between these two parts of the track. It would be helpful to see Figures 4c-e reproduced for Environment B. There are related plots in the supplement, but they are not identical to the measures in Figure 4c-e. The authors note that running speed or dwell time seems to be less critical in Environment B than Environment A. Although the model seems to reproduce the experimental findings and thus indicates that the proposed mechanisms could work, it was hard to understand the reasoning of what matters most for EC3's input in terms of driving BTSP events in CA1. In Figure 4, I thought it was the product of dwell time and dF/F , meaning overall number of EC3 events. However, the Figure 5 data seem to contradict this idea (it would be nice to see Figure 4e for Env B to make sure). I therefore struggled to understand what aspects of EC3 inputs to CA1 were determining the BTSP events and thus the reshaping of the place field distribution along the track. Again, I understand that the model can reproduce these findings, but a simple explanation or intuition was lacking and more explanation would be helpful.

3. The examples of the rapid emergence of place fields in Figure 2b are very striking. It would be helpful to see a quantification of the time course of place field emergence across all the cells with new place fields. Do they all emerge as rapidly as the three examples? Do some emerge more slowly? This point is critical to understand how prevalent this key prediction of BTSP is across the population. Are there three cherry-picked examples, or is this a general phenomenon across the entire population?

Referee #2 (Remarks to the Author):

In this study Grienberger and Magee obtained pyramidal cell activity in CA1 using two-photon calcium imaging in head-fixed mice exposed to a linear treadmill, on which they learned to find reward locations. Previous work from the same group showed that over-representation of spatial information causes 'behavioral timescale synaptic plasticity' (BTSP) in apical dendrites of CA1 pyramidal cells as important pre-requisite for the emergence of CA1 place cells. BTSP is based on calcium plateau potentials, which require the activation of presynaptic inputs from Layer III entorhinal cortex (EC3), activation of voltage gated calcium channels and NMDA receptors in the target cells. The data shown in this manuscript suggest that EC3 inputs provide an instructive signal to pyramidal cells directing BTSP to generate over-representation of reward locations. Since the effect of EC3 inputs on pyramidal cell BTSP was not overwhelmingly strong on the initially used linear track (environment A), they moved to another more enriched environment B. It was equipped

with a stronger and more prominent reward-predictive cue. The authors found that in environment B, EC3 activity in stratum lacunosum moleculare was more elevated (as depicted by 2-photon calcium imaging) close to cue location and CA1 place field density was higher close to the cue location than at the reward location, suggest that EC3 inputs remap between environments and may provide an instructive signal to CA1 pyramidal cells for the representation of relevant cues and in turn, for the execution of the behavior.

This is an elegant study providing information on changes in EC3 activity in dependence on the environmental setting and its behavioral relevance. Its clear in the content and message. The quality of the data is high. The robustness of the conclusions is not fully solid based on lacking interference approaches with EC inputs.

The criticism is as follows:

1. The imaging data indicate that EC input activity is evenly distributed across the linear track (line 116: df/f for all axons was basically constant across linear track spatial location). This finding is surprising given that previous studies showed significant spatial tuning and place field characteristics of EC layer III cells based on single unit recordings from EC cell somata in layer III of freely navigating rodents (Diehl et al., Neuron 2017) and based on calcium imaging data from GCAMP-loaded MEC axons in the stratum lacunosum moleculare (Cholvin et al., Neuron 2021). Moreover, both publications demonstrated that layer III EC inputs drastically change their preferred firing locations in mice navigating through different environments. Thus, the observation that EC inputs remap between environments has been provided. It should therefore be expected that changes in the attention related to the cues within an given environment will modulate EC input activity, which in turn will affect the emergence of BTSPs and pyramidal cell activity.

2. If running speed is a good predictive signal for the reward over-representation location in environment A but not in environment B, the latter being characterized by a strong reward predictive cue (light flash), than the question comes up whether BTSPs emerging under various behavioral conditions might be generated by different sets of cellular and network mechanisms. In other words do different external conditions evoke the same phenomenon (BTSP, emergence of place cells) on the basis of different mechanisms?

In this context, it remained unclear whether the postulated instructive signal provided by the EC3 is indeed required for the emergence of BTSP in environment B. One test could be to interfere with the EC input using optogenetic or pharmacogenetic tools. Moreover, does the instructive signal contains any information about space, context and cue? Is it indeed induced at certain attentive states of the animal?

3. The authors state that EC activity was equal across the track. The reviewer is wondering whether some kind of periodicity or spatial information content can be obtained from the data and whether indications of different cue content or spatial information content can be related to the EC axon activity. Moreover, how could the authors be sure that they don't record from one and the same cell in a multiplicative manner by for example imaging axon collateral's from the same cell? The quality

of calcium transients in Figure 3 are hard to see in order to have a good idea on the quality of EC axon recordings.

Referee #3 (Remarks to the Author):

In this manuscript, the authors performed 2-photon calcium imaging from hippocampal CA1 neurons and input fibers in the dendritic layers while mice were exposed to a new environment with reward at a fixed location. It was shown in previous studies that in such a condition, new CA1 place cells emerge at the reward location within a single session. A previous paper from the same group beautifully showed that this recruitment of place cells involves a ~second-timescale plasticity, termed behavioral timescale synaptic plasticity (BTSP) triggered by plateau potential (dendritic spikes) of pyramidal cells (Bittner et al., Science 2017). In this paper, author aimed to test their hypothesis that the place cell recruitment is mediated by inputs from the entorhinal cortex via BTSP when animals are learning new reward location. Although this hypothesis is significant to be tested, I have following concerns:

Major concerns:

1. Experimental evidence showing that BTSP occurs by the inputs from the entorhinal cortex is missing in the current form. A critical experiment would be to record plateau potentials while manipulating entorhinal inputs.
2. I have a concern on their method to label entorhinal inputs. Authors specifically focus on inputs from layer 3 of the entorhinal cortex. However, in Fig. 3a (entorhinal input inhibition) cell bodies look to be located in the subiculum. In Fig. 3d (entorhinal input calcium imaging) cell bodies look to be in layer 2-5 of the ventral medial entorhinal cortex. It is not clear from the results if their manipulation and recording were done specifically on layer 3 neurons in the entorhinal cortex.
3. Related to above, it was previously shown that dorsal medial entorhinal cortex has reward-representing neurons (Boccarda et al., Science 2019; Butler et al., Science 2019). With such significant evidence, a straightforward hypothesis would be that these entorhinal reward neurons direct place cell emergence. Although authors did not see any reward place representing inputs in Figs 3 and 4, this is presumably because their labeling in Fig. 3d is in the ventral medial entorhinal cortex.
4. The link between behavioral learning, place cell emergence and entorhinal inputs is obscure. As pointed in #2 above, the manipulation and recording look like to be performed from different population, making it difficult to understand such a link. Also, did the inhibition of entorhinal cortex affect the learning in Fig. 3a-c?

Author Rebuttals to Initial Comments:

Response to Referee's comments

We would like to thank the reviewers for their time and constructive comments. To address their concerns, we performed additional analyses, experiments and simulations that resulted in three new figures (new Figure 6, new Extended Data Figures 4 and 11) and the modification of Figures 4 and 5 and Extended Data Figures 8-9 (formerly 7-8).

Referee #1 (Remarks to the Author):

This paper studies the mechanisms of CA1 place code reorganization during learning, in particular for the enrichment of place cells around a reward location. The main finding is that behavioral timescale synaptic plasticity contributes to this reorganization and that BTSP in CA1 originates from EC3 synaptic inputs. I found the paper to be exciting in linking BTSP to a well known reshaping of the place code in the hippocampus, thus extending the previous findings from this group on the characterization of BTSP. I also found the mechanistic investigation of EC3's input to be exciting and important. I appreciated the combination of measurements of neural activity, manipulations of specific circuit elements, and computational modeling. Overall, I found the study to be well done and thorough and the paper to be well written and clearly presented. I had one comment that lessened my enthusiasm slightly, regarding the link of BTSP to the behavior, and a few specific comments.

We thank the referee for their positive comments.

1. The paper is introduced by talking about how learning underlies adaptive behavior and how learning of a reward site requires an over-representation of place cells near the reward site (first two sentences of the abstract). In this paper, disrupting the EC3 inputs to CA1 disrupts the over-representation near the reward site (Figure 3b-c). However, the behavior is not disrupted and appears similar to when EC3 inputs are intact (ED Figure 4e). This implies that BTSP can be disrupted and that the place cell reorganization can be prevented without any effect on the behavior.

This is a misunderstanding that likely results from us not having properly described the experiment. As with our pharmacological manipulations, we designed our optogenetic manipulation experiment to have as limited as possible an impact on EC activity and, therefore, behavior. We used a retrograde virus strategy (Tervo et al., 2016, Neuron) to limit ArchT expression specifically to the EC3 neurons that innervated the area of CA1 in which we were recording activity. Newly included histology shows that the expression of ArchT in EC3 was limited to a subregion of ipsilateral EC layer 3 neurons (Extended Data Fig. 4). Thus, the changes in CA1 population activity are, by design, limited to a small region of dorsal CA1 of a single hemisphere. The studies we cite on the role of the overrepresentation for learning (e.g., Turi et al., 2019, Neuron) used bilateral widespread expression and bilateral optogenetic manipulation of the hippocampal CA1, and aimed to impact a large fraction of CA1 population activity. Thus, we conclude that our optogenetic manipulations of EC were too limited in scale to alter the overall CA1 population activity sufficiently to affect the licking behavior of our mice. We have added text to make this explicit.

Related to this point, in the logic and model of Figure 4, the key determinant of the EC3 effect on place cell enrichment near the reward site seems to be the longer dwell time of the mouse near the reward site. If so, then the increase in dwell time near the reward site, which is a behavioral indicator of learning, precedes the BTSP events. Thus, the BTSP events might be causal for the place cell enrichment, but these events and the place cell enrichment would not be causal for the behavioral learning as they follow in time behaviors that are indicative of learning.

This is an interesting point. We generally view both the licking behavior and the running profile as indicators of learning. We find that there is a strong correlation between CA1 population activity changes and both the evolution of licking and running with trials. We include these plots below in Figure 1. We think this makes logical sense in that the environment must first be experienced before, or at least

simultaneous, as it is learned. We have not included this analysis in the manuscript because, as stated above, we have not directly tested the idea that experience-dependent changes in CA1 population activity are required for the evolution of licking behavior (as opposed to correlated with). Other labs have directly tested this idea.

Figure 1: CA1 population vector (PV) correlation, licking and running behaviors as a function of lap number. There is an inverse correlation between the fraction of licks made outside the reward zone or the velocity drop around the reward (both parameters indicate learning) and the development of the CA1 representation in environment A (top row; n=18 mice) and environment B (bottom row; n=9 mice).

2. Figure 4 was presented in a way that indicates that the dwell time was critical and that one could predict the BTSP events based on the product of the dwell time and dF/F , as presented in Figure 4e. However, this does not seem to be true for the experiments in Figure 5. The EC3 axons have higher activity at the blue light time and longer dwell times near the reward. I therefore anticipate that the product of dF/F and dwell time would be similar between these two parts of the track.

This comment by the referee and another has led us to dig deeper into the data. As a result, Figs. 4 and 5 as well as Extended Data Figs. 8-9 and the associated text have been substantially revised. The bottom-line is that we have uncovered a third element that is involved in shaping the CA1 representation. Thus, in addition to the spatial distribution of moderately tuned EC3 neurons and the running behavior, we found that the level of tuning (expressed as selectivity and activity correlation of odd. vs. even laps, i.e., stability; Fig. 4d) is enhanced in the neurons with peak firing around the reward location in environment A. Inclusion of this element into our models increased the accuracy of the models (Fig. 4h). So, there are three elements essential to the EC3 instructive signal: 1) the spatial distribution of EC3 inputs, 2) the spatial distribution of the tuning level of those inputs and 3) the running behavior of the mice. The switch from environment A to environment B with a predictive visual cue causes the number of EC3 neurons tuned to the cue location to increase ~ 4 fold and the neurons with enhanced tuning also shift to the cue (Extended Data Fig. 9f). In addition, the running behavior is quite different between the two environments (Figs. 4f, 5c, 5g). The running profile of animals in environment A shows a more distinctive or pronounced slowing only around the reward location. In contrast in environment B, the animals initially slow at the predictive light cue and then continue to slow until the reward location. So, to answer the question directly, our models suggest that the impact of the running is similar in both environments and can be observed as the difference between the predictions produced for constant and actual speeds (Figs. 4g and 5h). The question becomes, why then does it “seem” different? In environment A both the increased dwell time and the highly tuned neurons were located at the same position (at the reward site), but in

environment B the highly tuned neurons were at a different location (the cue site) than the regions with the longest dwell times (the reward site). The contribution of the additional factor (tuning level) along with the different running patterns in the two environments made it appear that in environment A the dwell time was more “critically” related to the CA1 overrepresentation than during environment B (where the tuning and the longest dwell time were at separate locations). We now realize that running is but one of three factors affecting the shape of the CA1 representation. We thank the reviewer for this comment and hope that we have now clarified this issue.

3. The examples of the rapid emergence of place fields in Figure 2b are very striking. It would be helpful to see a quantification of the time course of place field emergence across all the cells with new place fields. Do they all emerge as rapidly as the three examples? Do some emerge more slowly? This point is critical to understand how prevalent this key prediction of BTSP is across the population. Are there three cherry-picked examples, or is this a general phenomenon across the entire population?

We agree with the reviewer that this place cell appearance is very striking. As such, it is consistent with the BTSP-mediated place field formation, which requires only a single trial. We quantify the time course of place field emergence by showing 1) the cumulative distribution of place field onset laps as a function of session length (Fig. 2c), 2) the peak location of the place fields formed across 4 sections of the recording session (Fig. 2d), and 3) the number of laps it took to complete 33% and 90% of the CA1 representation (Extended Data Fig. 1). To identify the lap where a place field appeared, a neuron’s activity has to fulfill the following criteria: 1) a lap with a spatial bin with significant Ca^{2+} activity (greater than 3 * standard deviation of the baseline) in lap X. 2) presence of spatial bins with significant Ca^{2+} activity in 2 out of the 5 following laps (lap X+1 to lap X+6). That means that, by definition, a place field onset requires reliable significant activity. To address the referee’s comment and to extend our analyses of the time course, we add a quantification of the number of laps with significant activity before place field onset. In environment A, only 1.24 ± 0.07 laps per place cell had significant activity at the eventual place field location before the actual place field onset (mean onset in lap 22.6 ± 0.7); when we do the same analysis for environment B, we find 1.7 ± 0.1 laps per place cell with significant activity before place field onset (mean onset in lap 28.6 ± 1.1). All values represent mean \pm SEM. We conclude that place field emergence in almost completely silent cells is a consistent feature in our linear based treadmill task (e.g., cell 2 is active in lap 23) and, that the examples that we show in Figure 2 are representative. We include this quantification in the new version of the manuscript.

Referee #2 (Remarks to the Author):

This is an elegant study providing information on changes in EC3 activity in dependence on the environmental setting and its behavioral relevance. Its clear in the content and message. The quality of the data is high. The robustness of the conclusions is not fully solid based on lacking interference approaches with EC inputs.

We thank the referee for their positive comments.

The criticism is as follows: 1. The imaging data indicate that EC input activity is evenly distributed across the linear track (line 116: df/f for all axons was basically constant across linear track spatial location). This finding is surprising given that previous studies showed significant spatial tuning and place field characteristics of EC layer III cells based on single unit recordings from EC cell somata in layer III of freely navigating rodents (Diehl et al., Neuron 2017) and based on calcium imaging data from GCAMP-loaded MEC axons in the stratum lacunosum moleculare (Cholvin et al., Neuron 2021). Moreover, both publications demonstrated that layer III EC inputs drastically change their preferred firing locations in mice navigating through different environments. Thus, the observation that EC inputs remap between environments has been provided. It should therefore be expected that changes in the attention related to the cues within a given environment will modulate EC input activity, which in turn will affect the emergence of BTSPs and pyramidal cell activity.

Based on the referee's comments, we have expanded our discussion of the EC3 neuronal activity in our recording as well as those previously published.

2. If running speed is a good predictive signal for the reward over-representation location in environment A but not in environment B, the latter being characterized by a strong reward predictive cue (light flash), than the question comes up whether BTSPs emerging under various behavioral conditions might be generated by different sets of cellular and network mechanisms. In other words do different external conditions evoke the same phenomenon (BTSP, emergence of place cells) on the basis of different mechanisms?

This comment by the referee and another has led us to dig deeper into the data. As a result, Figs. 4 and 5 as well as Extended Data Figs. 8-9 and the associated text have been substantially revised. The bottom-line is that we have uncovered a third element that is involved in shaping the CA1 representation. Thus, in addition to the spatial distribution of moderately tuned EC3 neurons and the running behavior, we found that the level of tuning (expressed as selectivity and activity correlation of odd. vs. even laps, i.e., stability; Fig. 4d) is enhanced in the neurons with peak firing around the reward location in environment A. Inclusion of this element into our models increased the accuracy of the models (Fig. 4h). So, there are three elements essential to the EC3 instructive signal: 1) the spatial distribution of EC3 inputs, 2) the spatial distribution of the tuning level of those inputs and 3) the running behavior of the mice. The switch from environment A to environment B with a predictive visual cue causes the number of EC3 neurons tuned to the cue location to increase ~4fold and the neurons with enhanced tuning also shift to the cue (Extended Data Fig. 9f). In addition, the running behavior is quite different between the two environments (Figs. 4f, 5c, 5g). The running profile of animals in environment A shows a more distinctive or pronounced slowing only around the reward location. In contrast in environment B, the animals initially slow at the predictive light cue and then continue to slow until the reward location. So, to answer the question directly, our models suggest that the impact of the running is similar in both environments and can be observed as the difference between the predictions produced for constant and actual speeds (Figs. 4g and 5h). The question becomes, why then does it "seem" different? In environment A both the increased dwell time and the highly tuned neurons were located at the same position (at the reward site), but in environment B the highly tuned neurons were at a different location (the cue site) than the regions with the longest dwell times (the reward site). The contribution of the additional factor (tuning level) along with the different running patterns in the two environments made it appear that in environment A the

dwelling time was more “critically” related to the CA1 overrepresentation than during environment B (where the tuning and the longest dwelling time were at separate locations). We now realize that running is but one of three factors affecting the shape of the CA1 representation. We thank the reviewer for this comment and hope that we have now clarified this issue.

In this context, it remained unclear whether the postulated instructive signal provided by the EC3 is indeed required for the emergence of BSTP in environment B. One test could be to interfere with the EC input using optogenetic or pharmacogenetic tools. Moreover, does the instructive signal contain any information about space, context and cue? Is it indeed induced at certain attentive states of the animal?

Based on these comments and others, we have performed an additional optogenetic manipulation experiment to directly test the role of non-uniform EC3 input in shaping CA1 population activity in environment B. In this case, we inhibited EC3 activity around the predictive light cue such that it should reduce the elevated EC3 activity in this region and examined the impact on the CA1 representation. We observed that the density of CA1 place cells around the cue was not significantly elevated under these conditions compared to other locations (Extended Data Fig. 11). Thus, we have shown under four different conditions (2 different environments with two different optogenetic manipulations) that the EC3 activity is strongly environmentally dependent and that the CA1 population activity directly follows the EC3 activity profile (spatial distributions of tuned neuronal activity and their actual level of tuning) as well as the running behavior of the mice. We conclude that EC3 input is necessary and sufficient for directing experience-dependent shaping of CA1 representations in both environments.

3. The authors state that EC activity was equal across the track. The reviewer is wondering whether some kind of periodicity or spatial information content can be obtained from the data and whether indications of different cue content or spatial information content can be related to the EC axon activity. Moreover, how could the authors be sure that they don't record from one and the same cell in a multiplicative manner by for example imaging axon collateral's from the same cell? The quality of calcium transients in Figure 3 are hard to see in order to have a good idea on the quality of EC axon recordings.

We address the first question here along with the last two related questions from point #2 above. We show that EC3 activity does possess spatial selectivity (Figures 3-5, Extended Data Fig. 7a-c), and the distribution of these neurons is altered by the environment (context and cues; Figures 4 & 5; Extended Data Figs. 7 and 9). In addition, we now include data showing that the level of selectivity and stability varies as a function of position (Fig 4d; Extended Data Fig. 9f). We have not observed any periodicity, although our track is rather short (180 cm). Moreover, while we have not directly calculated spatial information on EC3 neuronal activity, we show in Extended Data Fig. 7d-f that the top 5% most selective neurons are approximately as selective as a standard place cell (and similar to that observed in the selected neurons presented in Cholvin et al., Neuron, 2021). So, the bottom-line is that there is a substantial amount of spatial tuning in many EC3 neurons, and this is quantified in several figures in the manuscript. Further, our models specifically contain transition probability manipulations to produce the desired level of spatial tuning in the “neurons” used in the simulations (Fig. 4 and Extended Data Figs. 7-9). Finally, we have not attempted to determine the role of attention in EC3 activity during learning, although this is a very exciting idea.

To answer this referee's second question, we first extracted axonal EC3 Ca²⁺ signals using the image registration and region of interest (ROI) detection algorithms of the Suite2P pipeline (Pachitariu et al., 2016, bioRxiv). Here, Suite2p relies on phase correlation to calculate the XY offset between any given frame and a reference image, which is followed by ROI detection as a group of pixels that are correlated across time. We then used the extracted fluorescent signal to perform a noise correlation analysis applying a Pearson correlation coefficient threshold of 0.4-0.5 to identify ROIs that likely originate from the same

axon/neuron, i.e., the activity of ROIs that crossed this threshold was combined for all the following analyses. This step-by-step procedure is shown in Extended Data Fig. 6a-d. This figure also includes Ca^{2+} transients from several axonal ROIs to illustrate the quality of our recordings. Please note that to ensure a successful motion correction, we co-expressed GCaMP6f and tdTomato in EC3 axons (Extended Data Fig. 6e-g). Only datasets, for which the motion correction was successful, were included in this study.

Referee #3 (Remarks to the Author):

In this manuscript, the authors performed 2-photon calcium imaging from hippocampal CA1 neurons and input fibers in the dendritic layers while mice were exposed to a new environment with reward at a fixed location. It was shown in previous studies that in such a condition, new CA1 place cells emerge at the reward location within a single session. A previous paper from the same group beautifully showed that this recruitment of place cells involves a ~second-timescale plasticity, termed behavioral timescale synaptic plasticity (BTSP) triggered by plateau potential (dendritic spikes) of pyramidal cells (Bittner et al., Science 2017). In this paper, author aimed to test their hypothesis that the place cell recruitment is mediated by inputs from the entorhinal cortex via BTSP when animals are learning new reward location. Although this hypothesis is significant to be tested, I have following concerns:

Major concerns:

1. Experimental evidence showing that BTSP occurs by the inputs from the entorhinal cortex is missing in the current form. A critical experiment would be to record plateau potentials while manipulating entorhinal inputs.

In the past we have shown that optogenetic inhibition of EC3 input reduces the probability of plateau potential initiation in CA1 cells recorded from mice under conditions similar to environment A (Bittner et al 2015). We have also presented data in the current manuscript, as well as in the past, that plateau potentials induce BTSP and place field formation (Bittner et al 2015, Grienberger et al 2017, Bittner et al 2017, Milstein et al 2021). We have also shown in the current manuscript that inhibition of EC3 input reduces place field formation and alters experience-dependent shaping of CA1 representations. Given all the current and past evidence we think it natural to conclude that EC3 activity drives plateau potentials in CA1 neurons to induce new place field formation via BTSP and that this is the primary mechanism by which learning-related changes in CA1 population activity occur.

2. I have a concern on their method to label entorhinal inputs. Authors specifically focus on inputs from layer 3 of the entorhinal cortex. However, in Fig. 3a (entorhinal input inhibition) cell bodies look to be located in the subiculum. In Fig. 3d (entorhinal input calcium imaging) cell bodies look to be in layer 2-5 of the ventral medial entorhinal cortex. It is not clear from the results if their manipulation and recording were done specifically on layer 3 neurons in the entorhinal cortex.

We thank the reviewer for bringing up this point. We apply two strategies to label EC3 neurons: 1) we use a retrograde (rg) viral strategy and co-inject AAVrg.syn.FLEX.ArchT.tdTomato (or tdTomato) and AAV2/1.syn.GCaMP6f unilaterally into dorsal CA1 of pOxr1-Cre mice (Cre recombinase is targeted to layer 3 neurons, Suh et al., 2011, Science). This approach allowed us to image dorsal CA1 neurons while specifically silencing those EC3 neurons that project to our recording area (Extended Data Fig. 4a-b). 2) We inject a mix of AAV2/1.FLEX.GCaMP6f and AAV2/1.FLEX.tdTomato into the EC. This approach allowed us to image axons in stratum lacunosum-moleculare of the ipsilateral dorsal CA1 region (Extended Data Fig. 4c-d). We now quantify the distribution of labeled somata in layer 3 EC and labeled axons of these EC3 neurons in the ipsilateral stratum lacunosum-moleculare of dorsal CA1 (Extended Data Fig. 4e). We find first that for both approaches, the somatic expression of fluorescent proteins is largely limited to EC layer 3 neurons and that we do not observe cell bodies in the subiculum. This conclusion is supported by our finding that the dentate gyrus is completely devoid of tdTomato fluorescence in case of the retrograde strategy (Extended Data Fig. 4b) and almost completely devoid of tdTomato fluorescence in case of the anterograde strategy (Extended Data Fig. 4d). Next, we find that the axons of the labeled layer 3 EC neurons cross layer 5 EC and the subiculum on their way to CA1. Finally, the retrograde viral strategy labeled a subset of the same EC3 population that the anterograde strategy labeled. This is by design as to have as limited as possible an impact on the overall EC activity. Importantly, regardless of the labeling approach used, we did record from or manipulate those EC3 axons that project to those dorsal CA1 neurons forming the overrepresentation.

3. Related to above, it was previously shown that dorsal medial entorhinal cortex has reward-representing neurons (Boccarda et al., Science 2019; Butler et al., Science 2019). With such significant evidence, a straightforward hypothesis would be that these entorhinal reward neurons direct place cell emergence. Although authors did not see any reward place representing inputs in Figs 3 and 4, this is presumably because their labeling in Fig. 3d is in the ventral medial entorhinal cortex.

We did find that some of our EC3 neurons were tuned to locations around the reward site (Figs. 4 & 5). These could be “reward neurons”, as described in the references mentioned by the reviewer. However, since we did not move the reward site, we cannot know if they are indeed tracking the reward or something else. It is also worth mentioning that neither of the two studies cited by the reviewer differentiates between layers 2 and 3. Therefore, it remains unclear whether these reward-representing neurons can be found in layer 3 EC or whether they are part of the layer 2 population. Finally, as mentioned above, we now include in the new version of the manuscript histological analyses (Extended Data Fig. 4), which show that, in our imaging experiments, we label layer 3 somata along the entire dorsal-ventral EC axis. However, our recording site is located in dorsal CA1, i.e., we focused our imaging (and manipulation) experiments on the axons of dorsal EC3 neurons providing input to those CA1 neurons forming the overrepresentations.

4. The link between behavioral learning, place cell emergence and entorhinal inputs is obscure. As pointed in #2 above, the manipulation and recording look like to be performed from different population, making it difficult to understand such a link. Also, did the inhibition of entorhinal cortex affect the learning in Fig. 3a-c?

This is a misunderstanding that likely results from our not having properly described the experiment. As with our pharmacological manipulations, the optogenetic manipulation experiment was specifically designed to have as prescribed as possible an impact on EC activity and, therefore behavior. We used a retrograde virus strategy to limit ArchT expression specifically to the EC3 neurons that innervated the area of CA1 in which we were recording activity. Newly included histology shows that the expression of ArchT in EC3 was limited to a subregion of ipsilateral EC layer 3 neurons (Extended Data Fig. 4). Thus, the changes in CA1 population activity are, by design, limited to a small region of CA1 of a single hemisphere. The cited studies performed in other labs, used widespread expression and opto-genetic manipulation in both sides of the dorsal hippocampus to impact a large fraction of CA1 population activity. Thus, we conclude that our optogenetic manipulations of EC were too limited in scale to alter the overall CA1 population activity enough to affect the licking behavior of our mice.

Reviewer Reports on the First Revision:

Referees' comments:

Referee #1 (Remarks to the Author):

The authors have thoroughly addressed my comments. The paper addresses an interesting question using rigorous and exciting approaches. I support publication. Congratulations on the great paper.

Referee #2 (Remarks to the Author):

I have no further comments or questions to the study. The response of the authors was clear.

Referee #3 (Remarks to the Author):

I appreciate the authors responding all of my major concerns. These responses were very helpful to digest their results. The addition of Fig E4 greatly helps histological evaluation.

I suggest including the following interpretations from their responses into the main text of the manuscript, so that readers can also follow their logics:

(Response to point 1): "In the past we have shown that optogenetic inhibition of EC3 input reduces the probability of plateau potential initiation in CA1 cells recorded from mice under conditions similar to environment A (Bittner et al 2015) ... Given all the current and past evidence we think it natural to conclude that EC3 activity drives plateau potentials in CA1 neurons to induce new place field formation via BTSP and that this is the primary mechanism by which learning-related changes in CA1 population activity occur."

(Response to point 4): "Thus, we conclude that our optogenetic manipulations of EC were too limited in scale to alter the overall CA1 population activity enough to affect the licking behavior of our mice."